# Zero-Regret Performative Prediction Under Inequality Constraints

**Wenjing Yan    Xuanyu Cao** *
Department of Electronic and Computer Engineering
The Hong Kong University of Science and Technology
`wj.yan@connect.ust.hk, eexcao@ust.hk`

## Abstract

Performative prediction is a recently proposed framework where predictions guide decision-making and hence influence future data distributions. Such performative phenomena are ubiquitous in various areas, such as transportation, finance, public policy, and recommendation systems. To date, work on performative prediction has only focused on unconstrained scenarios, neglecting the fact that many real-world learning problems are subject to constraints. This paper bridges this gap by studying performative prediction under inequality constraints. Unlike most existing work that provides only performative stable points, we aim to find the optimal solutions. Anticipating performative gradients is a challenging task, due to the agnostic performative effect on data distributions. To address this issue, we first develop a robust primal-dual framework that requires only approximate gradients up to a certain accuracy, yet delivers the same order of performance as the stochastic primal-dual algorithm without performativity. Based on this framework, we then propose an adaptive primal-dual algorithm for location family. Our analysis demonstrates that the proposed adaptive primal-dual algorithm attains $\mathcal{O}(\sqrt{T})$ regret and constraint violations, using only $\sqrt{T} + 2T$ samples, where $T$ is the time horizon. To our best knowledge, this is the first study and analysis on the optimality of the performative prediction problem under inequality constraints. Finally, we validate the effectiveness of our algorithm and theoretical results through numerical simulations.

## 1   Introduction

Stochastic optimization plays a critical role in statistical sciences and data-driven computing, where the goal is to learn decision rules (e.g., classifiers) based on limited samples that generalize well to the entire population. Most prior studies on stochastic optimization [Heyman and Sobel, 2004; Karimi et al., 2019; Powell, 2019] rely on the assumption that the data of the entire population follows a static distribution. This assumption, however, does not hold in applications where the data distributions change dynamically in response to decision-makers' actions [Hardt et al., 2016; Dong et al., 2018]. For instance, in transportation, travel time estimates [Mori et al., 2015] influence routing decisions, resulting in realized travel times; in banking, credit evaluation criteria [Abdou and Pointon, 2011] guide borrowers' behaviors and subsequently their credit scores; and in advertising, recommendations [García-Sánchez et al., 2020] shape customer preferences, leading to consumption patterns. Such interplay between decision-making and data distribution arises widely in various areas, such as transportation, finance, public policy, and recommendation systems.

The seminal work [Perdomo et al., 2020] formalized the phenomenon as *performative prediction*, which represents the strategic responses of data distributions to the taken decisions via decision-

---

*Corresponding Author.

37th Conference on Neural Information Processing Systems (NeurIPS 2023).

dependent distribution maps [Quinonero-Candela et al., 2008]. Since then, an increasing body of research has been dedicated to performative prediction problems. Most existing studies are focused on identifying *performative stable points* [Li and Wai, 2022; Li et al., 2022; Drusvyatskiy and Xiao, 2022; Brown et al., 2022; Mendler-Dünner et al., 2020; Wood et al., 2021; Ray et al., 2022], given the complexities of the decision-induced distribution shifts and the unknown decision-dependent distributions. The proposed algorithms typically iteratively retrain the deployed models until convergence. However, performative stability generally does not imply performative optimality. Aiming to achieve optimal performance, a few recent works designed effective algorithms by leveraging rich performative feedback [Jagadeesan et al., 2022], or by making some parametric assumptions on the underlying distribution maps. For instance, the distribution maps belong to the location family with linear structure [Miller et al., 2021] or are from the exponential family [Izzo et al., 2021].

All the aforementioned work on performative prediction is focused on unconstrained learning problems. However, in the real world, many performative prediction applications are subject to constraints [Detassis et al., 2021; Wood and Dall'Anese, 2022a]. Constraints can be used to ensure the satisfaction of desired properties, such as fairness, safety, and diversity. Examples include safety and efficiency constraints in transportation [Metz, 2021], relevance and diversity constraints in advertising [Khamis, 2020], and risk tolerance and portfolio constraints in financial trading [Föllmer and Schied, 2002], etc. In addition, constraints can serve as side information to enhance the learning outcomes, e.g., by narrowing the scope of exploration or by incorporating prior knowledge Serafini and Garcez [2016]; Wu et al. [2018]. As performative shifts can rarely be analyzed offline, incorporating constraints on what constitutes safe exploration [Turchetta et al., 2019] or facilitates optimization [Wood and Dall'Anese, 2022b] is of crucial importance.

Despite its importance, research on performative prediction under constraints has been neglected so far. Although some work [Izzo et al., 2021; Piliouras and Yu, 2022] restricted decision variables to certain regions, this feasible set restriction was simply handled by projections. This paper bridges this gap by studying the performative prediction problem under inequality constraints, for which simple projection is inadequate to handle. Unlike most existing work that provides only performative stable points, we aim to find the optimal solutions. As aforementioned, finding performative optima is a challenging task because we now need to anticipate performative effect actively rather than simply retrain models in a myopic manner.

However, the performative effect on distribution maps is unknown, which hinders the computation of exact performative gradient. To solve this problem, we develop a robust primal-dual framework that admits inexact gradients. We ask the following questions: *How does the gradient approximation error affect the performance of the primal-dual framework? Under what accuracy can the approximate gradients maintain the performance order of the stochastic primal-dual algorithm without performativity? How to construct effective gradient approximations that attain the desired accuracy?* We answer the above thoroughly. Our idea hinges on enhancing gradient approximation with the parametric knowledge of distribution maps. In particular, we follow existing studies [Miller et al., 2021; Jagadeesan et al., 2022] and focus on the family of location maps. Location family exhibit a favorable linear structure for algorithm development while maintaining broad generality to model many real-world applications. Distribution maps of this type are ubiquitous throughout the performative prediction literature, such as strategic classification [Hardt et al., 2016; Perdomo et al., 2020], linear regression [Miller et al., 2021], email spam classification [Li et al., 2022], ride-share [Narang et al., 2022], among others. Nevertheless, we emphasize that our robust primal-dual framework is applicable to other forms of distributions with effective gradient approximation methods.

To our best knowledge, this paper provides the first study and analysis on the optimality of performative prediction problems under inequality constraints. We highlight the following key contributions:

- We develop a robust primal-dual framework that requires only approximate gradients up to an accuracy of $\mathcal{O}(\sqrt{T})$, yet delivers the same order of performance as the stochastic primal-dual algorithm without performativity, where $T$ is the time horizon. Notably, the robust primal-dual framework does not restrict the approximate gradients to be unbiased and hence offers more flexibility to the design of gradient approximation.

- Based on this framework, we propose an adaptive primal-dual algorithm for location family, which consists of an online stochastic approximation and an offline parameter estimation for the performative gradient approximation. Our analysis demonstrates that the proposed algorithm achieves $\mathcal{O}(\sqrt{T})$ regret and constraint violations, using only $\sqrt{T} + 2T$ samples.

Finally, we conduct experiments on two examples: multi-task linear regression and multi-asset portfolio. The numerical results validate the effectiveness of our algorithm and theoretical analysis.

## 1.1 Related Work

The study on performative prediction was initiated in [Perdomo et al., 2020], where the authors defined the notion of performative stability and demonstrated how performative stable points can be found through repeated risk minimization and stochastic gradient methods. Since then, substantial efforts have been dedicated to identifying performative stable points in various settings, such as single-agent [Mendler-Dünner et al., 2020; Drusvyatskiy and Xiao, 2022; Brown et al., 2022], multi-agent [Li et al., 2022; Piliouras and Yu, 2022], games [Narang et al., 2022], reinforcement learning [Mandal et al., 2023], and online learning [Wood et al., 2021; Wood and Dall'Anese, 2022a].

A few recent works aimed to achieve performative optimality, a more stringent solution concept than performative stability. In [Miller et al., 2021], the authors evaluated the conditions under which the performative problem is convex and proposed a two-stage algorithm to find the performative optima for distribution maps in location family. Another paper on performative optimality is [Izzo et al., 2021], which proposed a PerfGD algorithm by exploiting the exponential structure of the underlying distribution maps. Both works took advantage of parametric assumptions on the distribution maps. Alternatively, [Jagadeesan et al., 2022] proposed a performative confidence bounds algorithm by leveraging rich performative feedback, where the key idea is to exhaustively explore the feasible region with an efficient discarding mechanism.

A closely related work is [Wood and Dall'Anese, 2022b], which studied stochastic saddle-point problems with decision-dependent distributions. The paper focused on performative stable points (equilibrium points), whereas we aim at the performative optima, which is more challenging. Another difference is that [Wood and Dall'Anese, 2022b] only demonstrated the convergence of the proposed primal-dual algorithm in the limit, without providing an explicit finite-time convergence rate. In contrast, we provide $\mathcal{O}(\sqrt{T})$ regret and $\mathcal{O}(\sqrt{T})$ constraint violation bounds for the proposed algorithm in this paper.

## 2 Problem Setup

We study a performative prediction problem with loss function $\ell(\boldsymbol{\theta}; Z)$, where $\boldsymbol{\theta} \in \boldsymbol{\Theta}$ is the decision variable, $Z \in \mathbb{R}^k$ is an instance, and $\boldsymbol{\Theta} \in \mathbb{R}^d$ is the set of available decisions. Different from in stationary stochastic optimization where distributions of instances are fixed, in performative prediction, the distribution of $Z$ varies with the decision variable $\boldsymbol{\theta}$, represented by $Z \sim \mathcal{D}(\boldsymbol{\theta})$. In this paper, we consider that the decision variable $\boldsymbol{\theta}$ is subject to a constraint $\mathbf{g}(\boldsymbol{\theta}) \preceq \mathbf{0}$, where $\mathbf{g}(\cdot) : \boldsymbol{\Theta} \to \mathbb{R}^m$. The constraint $\mathbf{g}(\cdot)$ can be imposed on $\boldsymbol{\theta}$ to ensure certain properties, such as fairness, safety, and diversity, or to incorporate prior knowledge. We assume that $\mathbf{g}(\cdot)$ is available at the decision-maker in advance of the optimization. Ideally, the goal of the decision-maker is to solve the following stochastic problem:

$$\min_{\boldsymbol{\theta} \in \boldsymbol{\Theta}} \quad \mathbb{E}_{Z \sim \mathcal{D}(\boldsymbol{\theta})} \ell(\boldsymbol{\theta}; Z) \quad \text{s.t.} \quad \mathbf{g}(\boldsymbol{\theta}) \preceq \mathbf{0}, \tag{1}$$

where $\mathbb{E}_{Z \sim \mathcal{D}(\boldsymbol{\theta})} \ell(\boldsymbol{\theta}; Z)$ is referred to as *performative risk*, denoted by $\mathrm{PR}(\boldsymbol{\theta})$.

Problem (1) is, however, impossible to be solved offline because the distribution map $\mathcal{D}(\boldsymbol{\theta})$ is unknown. Instead, the decision-maker needs to interact with the environment by making decisions to explore the underlying distributions. Given the online nature of this task, we measure the loss of a sequence of chosen decisions $\boldsymbol{\theta}_1, \cdots, \boldsymbol{\theta}_T$ by *performative regret*, defined as

$$\mathrm{Reg}(T) := \sum_{t=1}^{T} \left( \mathbb{E}[\mathrm{PR}(\boldsymbol{\theta}_t)] - \mathrm{PR}(\boldsymbol{\theta}_{\mathrm{PO}}) \right),$$

where the expectation is taken over the possible randomness in the choice of $\{\boldsymbol{\theta}_t\}_{t=1}^{T}$, and $\boldsymbol{\theta}_{\mathrm{PO}}$ is the performative optimum, defined as

$$\boldsymbol{\theta}_{\mathrm{PO}} \in \arg\min_{\boldsymbol{\theta} \in \boldsymbol{\Theta}} \quad \mathbb{E}_{Z \sim \mathcal{D}(\boldsymbol{\theta})} \ell(\boldsymbol{\theta}; Z) \quad \text{s.t.} \quad \mathbf{g}(\boldsymbol{\theta}) \preceq \mathbf{0}.$$

Performative regret measures the suboptimality of the chosen decisions relative to the performative optima. Another performance metric for problem (1) on evaluating the decision sequence $\{\boldsymbol{\theta}_t\}_{t=1}^{T}$ is

*constraint violation*, given by

$$\mathrm{Vio}_i(T) := \sum_{t=1}^{T} \mathbb{E}[g_i(\boldsymbol{\theta}_t)], \forall i \in [m],$$

where we use the symbol $[m]$ to represent the integer set $\{1, \cdots, m\}$ throughout this paper.

Applications pertaining to problem (1) are ubiquitous. Next is an example.

**Example 1** (**Multi-Asset Portfolio**). *Consider a scenario where an investor wants to allocate his/her investment across a set of $l$ assets, such as stocks, bonds, and commodities. The objective is to maximize the expected return subject to certain constraints, including liquidity, diversity, and risk tolerance. Let $z_i$ denote the rate of return of the $i$th asset and $\theta_i$ denote its weight of allocation, $\forall i \in [l]$. The investment can affect the future rates of return of the assets and, consequently, the overall expected return of the portfolio. For example, excessive investment in a particular asset may lead to a decline in the rate of return of other assets. Let $\mathbf{z} = [z_1, \cdots, z_l]^\top$ and $\boldsymbol{\theta} = [\theta_1, \cdots, \theta_l]^\top$. Then, the expected return of the portfolio is $\mathbb{E}[r_p] := \mathbb{E}_{\mathbf{z} \sim \mathcal{D}(\boldsymbol{\theta})} \mathbf{z}^\top \boldsymbol{\theta}$. Typically, the risk of the portfolio is measured by the variance of its returns, given by $\boldsymbol{\theta}^\mathrm{T} \boldsymbol{\Psi} \boldsymbol{\theta}$, where $\boldsymbol{\Psi}$ is the covariance matrix of $\mathbf{z}$. One common approach to model liquidity is using the bid-ask spread, which measures the gap between the highest price a buyer is willing to pay (the bid) and the lowest price a seller is willing to accept (the ask) for a particular asset. Denote the vector of the bid-ask spread of the $l$ assets by $\mathbf{s} = [s_1, \cdots, s_l]^\top$. Then, a liquidity constraint on the portfolio can be defined as $\mathbf{s}^\top \boldsymbol{\theta} \leq S$, where $S$ is the maximum allowable bid-ask spread. The multi-asset portfolio problem can be formulated as:*

$$\min_{\boldsymbol{\theta}} \ -\mathbb{E}_{\mathbf{z} \sim \mathcal{D}(\boldsymbol{\theta})} \mathbf{z}^\top \boldsymbol{\theta} \quad \text{s.t.} \ \sum_{i=1}^{l} \theta_i \leq 1, \ \mathbf{0} \preceq \boldsymbol{\theta} \preceq \epsilon \cdot \mathbf{1}, \ \mathbf{s}^\top \boldsymbol{\theta} \leq S, \ \text{and} \ \boldsymbol{\theta}^\mathrm{T} \boldsymbol{\Psi} \boldsymbol{\theta} \leq \rho,$$

*where $\epsilon$ restricts the maximum amount of investment to one asset, and $\rho$ is the risk tolerance threshold.*

In this paper, our goal is to design an online algorithm that achieves both sublinear regret and sublinear constraint violations with respect to the time horizon $T$, i.e., $\mathrm{Reg}(T) \leq o(T)$ and $\mathrm{Vio}_i(T) \leq o(T)$, for all $i \in [m]$. Then, the time-average regret satisfies $\mathrm{Reg}(T)/T \leq o(1)$, and the time-average constraint violations satisfy $\mathrm{Vio}_i(T)/T \leq o(1)$, for all $i \in [m]$. Both asymptotically go to zero as $T$ goes to infinity. Therefore, the performance of the decision sequence $\{\boldsymbol{\theta}_t\}_{t=1}^{T}$ generated by the algorithm approaches that of the performative optimum $\boldsymbol{\theta}_{\mathrm{PO}}$ as $T$ goes to infinity.

## 3 Adaptive Primal-Dual Algorithm

### 3.1 Robust Primal-Dual Framework

In this subsection, we develop a robust primal-dual framework for the performative prediction problem under inequality constraints. Our approach involves finding a saddle point for the regularized Lagrangian of problem (1). The Lagrangian, denoted by $\mathcal{L}(\boldsymbol{\theta}, \boldsymbol{\lambda})$, is defined as

$$\mathcal{L}(\boldsymbol{\theta}, \boldsymbol{\lambda}) := \mathrm{PR}(\boldsymbol{\theta}) + \boldsymbol{\lambda}^\top \mathbf{g}(\boldsymbol{\theta}) - \frac{\delta\eta}{2} \|\boldsymbol{\lambda}\|_2^2, \tag{2}$$

where $\boldsymbol{\theta}$ is the primal variable (decision), $\boldsymbol{\lambda}$ is the dual variable (multiplier), $\eta > 0$ is the stepsize of the algorithm, and $\delta > 0$ is a control parameter. In (2), we add the regularizer $-\frac{\delta\eta}{2} \|\boldsymbol{\lambda}\|_2^2$ to suppress the growth of the multiplier $\boldsymbol{\lambda}$, so as to improve the stability of the algorithm.

To find the saddle point of the Lagrangian $\mathcal{L}(\boldsymbol{\theta}, \boldsymbol{\lambda})$, we utilize alternating gradient update on the primal variable $\boldsymbol{\theta}$ and the dual variable $\boldsymbol{\lambda}$. The gradients of $\mathcal{L}(\boldsymbol{\theta}, \boldsymbol{\lambda})$ with respect to $\boldsymbol{\theta}$ and $\boldsymbol{\lambda}$ are respectively given by

$$\nabla_{\boldsymbol{\theta}} \mathcal{L}(\boldsymbol{\theta}, \boldsymbol{\lambda}) = \nabla_{\boldsymbol{\theta}} \mathrm{PR}(\boldsymbol{\theta}) + \nabla_{\boldsymbol{\theta}} \mathbf{g}(\boldsymbol{\theta})^\top \boldsymbol{\lambda}, \tag{3}$$
$$\nabla_{\boldsymbol{\lambda}} \mathcal{L}(\boldsymbol{\theta}, \boldsymbol{\lambda}) = \mathbf{g}(\boldsymbol{\theta}) - \delta\eta \boldsymbol{\lambda},$$

where $\nabla_{\boldsymbol{\theta}} \mathbf{g}(\boldsymbol{\theta})$ is the Jacobian matrix of $\mathbf{g}(\cdot)$. In (3), $\nabla_{\boldsymbol{\theta}} \mathrm{PR}(\boldsymbol{\theta})$ is the gradient of the performative risk $\mathrm{PR}(\boldsymbol{\theta})$, given by

$$\nabla_{\boldsymbol{\theta}} \mathrm{PR}(\boldsymbol{\theta}) = \mathbb{E}_{Z \sim \mathcal{D}(\boldsymbol{\theta})} \nabla_{\boldsymbol{\theta}} \ell(\boldsymbol{\theta}; Z) + \mathbb{E}_{Z \sim \mathcal{D}(\boldsymbol{\theta})} \ell(\boldsymbol{\theta}; Z) \nabla_{\boldsymbol{\theta}} \log p_{\boldsymbol{\theta}}(Z), \tag{4}$$

where $p_{\boldsymbol{\theta}}(Z)$ is the density of $\mathcal{D}(\boldsymbol{\theta})$.

Since the data distribution $\mathcal{D}(\boldsymbol{\theta})$ is unknown, the exact gradient of the performative risk $\mathrm{PR}(\boldsymbol{\theta})$ is unavailable, posing a significant challenge to the algorithm design. In this paper, we tackle this

issue using a robust primal-dual framework. The main idea is to construct gradient approximations from data and then perform alternating gradient updates based on the inexact gradients. Denote by $\nabla_{\boldsymbol{\theta}}\widehat{\mathrm{PR}}_t(\boldsymbol{\theta})$ the approximation of the gradient $\nabla_{\boldsymbol{\theta}}\mathrm{PR}(\boldsymbol{\theta})$ at the $t$th iteration. Correspondingly, an approximation for the Lagrangian gradient $\nabla_{\boldsymbol{\theta}}\mathcal{L}(\boldsymbol{\theta}, \boldsymbol{\lambda})$ at the $t$th iteration is given by

$$\nabla_{\boldsymbol{\theta}}\widehat{\mathcal{L}}_t(\boldsymbol{\theta}, \boldsymbol{\lambda}) := \nabla_{\boldsymbol{\theta}}\widehat{\mathrm{PR}}_t(\boldsymbol{\theta}) + \nabla_{\boldsymbol{\theta}}\mathbf{g}(\boldsymbol{\theta})^{\top}\boldsymbol{\lambda}, \forall t \in [T].$$

The robust alternating gradient update is then performed as

$$\boldsymbol{\theta}_{t+1} = \Pi_{\boldsymbol{\Theta}}\left(\boldsymbol{\theta}_t - \eta\nabla_{\boldsymbol{\theta}}\widehat{\mathcal{L}}_t(\boldsymbol{\theta}_t, \boldsymbol{\lambda}_t)\right), \tag{5}$$

$$\boldsymbol{\lambda}_{t+1} = [\boldsymbol{\lambda}_t + \eta\nabla_{\boldsymbol{\lambda}}\mathcal{L}_t(\boldsymbol{\theta}_t, \boldsymbol{\lambda}_t)]^{+}. \tag{6}$$

Then, the next question is how to construct effective gradient approximations that achieve satisfactory performance.

By (4), the expectation over $\mathcal{D}(\boldsymbol{\theta})$ in the gradient $\nabla_{\boldsymbol{\theta}}\mathrm{PR}(\boldsymbol{\theta})$ can be approximated by samples, while the unknown probability density $p_{\boldsymbol{\theta}}(Z)$ presents the main challenge. Most existing research circumvented this problem by omitting the second term in $\nabla_{\boldsymbol{\theta}}\mathrm{PR}(\boldsymbol{\theta})$. This essentially gives a performative stable point. However, as pointed out in [Miller et al., 2021], performative stable points can be arbitrarily sub-optimal, leading to vacuous solutions. Instead, if we have further knowledge about the parametric structure of $p_{\boldsymbol{\theta}}(Z)$, the complexity of gradient approximation can be greatly reduced. In this regard, [Miller et al., 2021] and [Jagadeesan et al., 2022] exploited the linear structure of location family, and [Izzo et al., 2021] considered distribution maps within exponential family. Following [Miller et al., 2021; Jagadeesan et al., 2022], we focus on the family of location maps in this paper because it exhibits a favorable linear structure for algorithm development while maintaining broad generality to various applications. Next, we develop an adaptive algorithm for problem (1) with location family distribution maps based on the above robust primal-dual framework.

### 3.2 Algorithm Design for Location family

In the setting of location family, the distribution map depends on $\boldsymbol{\theta}$ via a linear shift, i.e.

$$Z \sim \mathcal{D}(\boldsymbol{\theta}) \Leftrightarrow Z \overset{d}{=} Z_0 + \mathbf{A}\boldsymbol{\theta}, \tag{7}$$

where $Z_0 \sim \mathcal{D}_0$ is a base component representing the data without performativity, $\mathbf{A} \in \mathbb{R}^{k \times d}$ captures the performative effect of decisions, and $\overset{d}{=}$ means equal in distribution. Denote by $\boldsymbol{\Sigma}$ the covariance matrix of the base distribution $\mathcal{D}_0$. Note that $\mathcal{D}_0$ is still unknown. Plugging the distribution definition (7) into (4), we obtain a more explicit expression for $\nabla_{\boldsymbol{\theta}}\mathrm{PR}(\boldsymbol{\theta})$ as

$$\nabla_{\boldsymbol{\theta}}\mathrm{PR}(\boldsymbol{\theta}) = \mathbb{E}_{Z_0 \sim \mathcal{D}_0}\left[\nabla_{\boldsymbol{\theta}}\ell\left(\boldsymbol{\theta}; Z_0 + \mathbf{A}\boldsymbol{\theta}\right) + \mathbf{A}^{\top}\nabla_Z\ell\left(\boldsymbol{\theta}; Z_0 + \mathbf{A}\boldsymbol{\theta}\right)\right].$$

To compute $\nabla_{\boldsymbol{\theta}}\mathrm{PR}(\boldsymbol{\theta})$, we still need to address two problems: the unknown base distribution $\mathcal{D}_0$ and the unknown performative parameter $\mathbf{A}$. We tackle them as follows.

**Offline Stochastic Approximation:** We approximate the base distribution $\mathcal{D}_0$ offline by sample average approximation [Kleywegt et al., 2002]. Specifically, before the start of the alternating gradient update, we first draw $n$ samples $\{Z_{0,i}\}_{i=1}^n$ from $\mathcal{D}(\mathbf{0})$. These samples are used to approximate the expectation over $Z_0$ throughout the algorithm iteration. Hence, the sample complexity from this expectation approximation is fixed at $n$.

**Online Parameter Estimation:** We estimate the parameter $\mathbf{A}$ via online least squares. In each round of the alternating gradient update, we first take the current decision $\boldsymbol{\theta}_t$ and its perturbed point $\boldsymbol{\theta}_t + \mathbf{u}_t$ to observe samples $Z_t \sim \mathcal{D}\left(\boldsymbol{\theta}_t\right)$ and $Z_t' \sim \mathcal{D}\left(\boldsymbol{\theta}_t + \mathbf{u}_t\right)$, respectively, where $\mathbf{u}_t$ is an injected noise specified by the decision-maker. We have $\mathbb{E}[Z_t - Z_t'|\mathbf{u}_t] = \mathbf{A}\mathbf{u}_t$. Then, the least-square problem at the $t$th iteration is designed as

$$\min_{\mathbf{A}} \frac{1}{2}\left\|Z_t' - Z_t - \mathbf{A}\mathbf{u}_t\right\|_2^2.$$

Let $\widehat{\mathbf{A}}_{t-1}$ be the estimate of $\mathbf{A}$ at the $(t-1)$th iteration. Based on it, we construct a new estimate $\widehat{\mathbf{A}}_t$ for $\mathbf{A}$ by using gradient descent on the above least-square objective. This gives us the update

$$\widehat{\mathbf{A}}_t = \widehat{\mathbf{A}}_{t-1} + \zeta_t\left(Z_t' - Z_t - \widehat{\mathbf{A}}_{t-1}\mathbf{u}_t\right)\mathbf{u}_t^{\top},$$

---

**Algorithm 1** Adaptive Primal-Dual Algorithm

---

1: Take decision $\boldsymbol{\theta} = \mathbf{0}$ and observe $n$ samples $Z_{0,i} \sim \mathcal{D}_0, \forall i \in [n]$.
2: Initialize $\boldsymbol{\theta}_1 \in \boldsymbol{\Theta}$ arbitrarily. Set $\boldsymbol{\lambda}_1 = \mathbf{0}$ and $\widehat{\mathbf{A}}_0 = \mathbf{0}$.
3: **for** $t = 1$ to $T$ **do**
4:      Take decision $\boldsymbol{\theta}_t$ and observe $Z_t \sim \mathcal{D}(\boldsymbol{\theta}_t)$.
5:      Generate noise $\mathbf{u}_t$.
6:      Take decision $\boldsymbol{\theta}_t + \mathbf{u}_t$ and observe $Z'_t \sim \mathcal{D}(\boldsymbol{\theta}_t + \mathbf{u}_t)$.
7:      Update parameter estimate by $\widehat{\mathbf{A}}_t = \widehat{\mathbf{A}}_{t-1} + \zeta_t \left( Z'_t - Z_t - \widehat{\mathbf{A}}_{t-1} \mathbf{u}_t \right) \mathbf{u}_t^\top$.
8:      Update gradient approximation $\nabla_{\boldsymbol{\theta}} \widehat{\mathrm{PR}}_t(\boldsymbol{\theta}_t)$ by (8).
9:      Compute $\nabla_{\boldsymbol{\theta}} \widehat{\mathcal{L}}_t(\boldsymbol{\theta}_t, \boldsymbol{\lambda}_t) = \nabla_{\boldsymbol{\theta}} \widehat{\mathrm{PR}}_t(\boldsymbol{\theta}_t) + \nabla_{\boldsymbol{\theta}} \mathbf{g}(\boldsymbol{\theta}_t)^\top \boldsymbol{\lambda}_t$.
10:      Update the primal variable by $\boldsymbol{\theta}_{t+1} = \Pi_{\boldsymbol{\Theta}} \left( \boldsymbol{\theta}_t - \eta \nabla_{\boldsymbol{\theta}} \widehat{\mathcal{L}}_t(\boldsymbol{\theta}_t, \boldsymbol{\lambda}_t) \right)$.
11:      Compute $\nabla_{\boldsymbol{\lambda}} \mathcal{L}(\boldsymbol{\theta}_t, \boldsymbol{\lambda}_t) = \mathbf{g}(\boldsymbol{\theta}_t) - \delta\eta\boldsymbol{\lambda}_t$.
12:      Update the dual variable by $\boldsymbol{\lambda}_{t+1} = [\boldsymbol{\lambda}_t + \eta \nabla_{\boldsymbol{\lambda}} \mathcal{L}(\boldsymbol{\theta}_t, \boldsymbol{\lambda}_t)]^+$.
13: **end for**

---

where $\zeta_t$ is the stepsize of the online least squares at the $t$th iteration.

**Adaptive Primal-Dual Algorithm:** With the above preparation, we obtain an approximation for the gradient $\nabla_{\boldsymbol{\theta}} \mathrm{PR}(\boldsymbol{\theta}_t)$ at the $t$th iteration as

$$\nabla_{\boldsymbol{\theta}} \widehat{\mathrm{PR}}_t(\boldsymbol{\theta}_t) := \frac{1}{n} \sum_{i=1}^n \left[ \nabla_{\boldsymbol{\theta}} \ell \left( \boldsymbol{\theta}_t; Z_{0,i} + \widehat{\mathbf{A}}_t \boldsymbol{\theta}_t \right) + \widehat{\mathbf{A}}_t^\top \nabla_Z \ell \left( \boldsymbol{\theta}_t; Z_{0,i} + \widehat{\mathbf{A}}_t \boldsymbol{\theta}_t \right) \right]. \tag{8}$$

Given $\nabla_{\boldsymbol{\theta}} \widehat{\mathrm{PR}}_t(\boldsymbol{\theta}_t)$, we develop an adaptive primal-dual algorithm for the constrained performative prediction problem (1) based on the robust primal-dual framework in § 3.1, which is presented in Algorithm 1. In Algorithm 1, the initial decision is randomly chosen from the admissible set $\boldsymbol{\Theta}$. Both the dual variable and the parameter estimate $\widehat{\mathbf{A}}_0$ are initialized to be zero. The algorithm maintains two sequences. One is the estimate $\widehat{\mathbf{A}}_t$, which is updated based on the newly observed samples $Z_t$ and $Z'_t$, as given in Step 7. The other is the alternating gradient update on the primal and dual variables, which are respectively given in Step 10 and Step 12.

**Remark 1.** *While this paper considers the distribution maps within the location family, we emphasize that the proposed robust primal-dual framework does not restrict to any form of distribution. For instance, the exponential family considered in [Izzo et al., 2021] with their gradient approximation method can be directly applied to our robust primal-dual framework.*

## 4 Convergence Analysis

In this section, we analyze the convergence performance of the proposed adaptive primal-dual algorithm. We first provide the convergence result of the robust primal-dual framework. Then, we bound the error of gradient approximation in our adaptive algorithm for the location family. With these results, the convergence bounds of the adaptive primal-dual algorithm are derived. Our analysis is based on the following assumptions.

**Assumption 1** (**Properties of** $\ell(\boldsymbol{\theta}; Z)$). *The loss function $\ell(\boldsymbol{\theta}; Z)$ is $\beta$-smooth, $L_{\boldsymbol{\theta}}$-Lipschitz continuous in $\boldsymbol{\theta}$, $L_Z$-Lipschitz continuous in $Z$, $\gamma_{\boldsymbol{\theta}}$-strongly convex in $\boldsymbol{\theta}$, and $\gamma_Z$-strongly convex in $Z$. Moreover, we have $\gamma_{\boldsymbol{\theta}} - \beta^2/\gamma_Z > 0$.*

**Assumption 2** (**Compactness and Boundedness of** $\boldsymbol{\Theta}$). *The set of admissible decisions $\boldsymbol{\Theta}$ is closed, convex, and bounded, i.e., there exists a constant $R > 0$ such that $\|\boldsymbol{\theta}\|_2 \le R, \forall \boldsymbol{\theta} \in \boldsymbol{\Theta}$.*

**Assumption 3** (**Properties of** $\mathbf{g}(\boldsymbol{\theta})$). *The constraint function $\mathbf{g}(\boldsymbol{\theta})$ is convex, $L_{\mathbf{g}}$-Lipschitz continuous, and bounded, i.e., there exists a constant $C$ such that $\|\mathbf{g}(\boldsymbol{\theta})\|_2 \le C, \forall \boldsymbol{\theta} \in \boldsymbol{\Theta}$.*

**Assumption 4** (**Bounded Stochastic Gradient Variance**). *For any $i \in [n]$ and $\boldsymbol{\theta} \in \boldsymbol{\Theta}$, there exists $\sigma \ge 0$ such that*

$$\mathbb{E}_{Z_{0,i} \sim \mathcal{D}_0} \left\| \nabla_{\boldsymbol{\theta}} \ell(\boldsymbol{\theta}; Z_{0,i} + \mathbf{A}\boldsymbol{\theta}) + \mathbf{A}^\top \nabla_Z \ell(\boldsymbol{\theta}; Z_{0,i} + \mathbf{A}\boldsymbol{\theta}) - \nabla_{\boldsymbol{\theta}} \mathrm{PR}(\boldsymbol{\theta}) \right\|_2^2 \le \sigma^2.$$

Assumption 1 is standard in the literature of performative prediction. Assumptions 2 and 3 are widely used in the analysis of constrained optimization problems [Tan et al., 2018; Yan et al., 2019; Cao and Başar, 2020], even with perfect knowledge of objectives. Assumption 4 bounds the variance of the stochastic gradient of $\mathrm{PR}(\boldsymbol{\theta})$. Additionally, to ensure a sufficient exploration of the parameter space, we make the following assumption on the injected noises $\{\mathbf{u}_t\}_{t=1}^{T}$.

**Assumption 5 (Injected Noise).** *The injected noises $\{\mathbf{u}_t\}_{t=1}^{T}$ are independent and identically distributed. Moreover, there exist positive constants $\kappa_1$, $\kappa_2$, and $\kappa_3$ such that for any $t \in [T]$, the random noise $\mathbf{u}_t$ satisfies*

$$\mathbf{0} \prec \kappa_1 \cdot \mathbf{I} \preceq \mathbb{E}\left[\mathbf{u}_t \mathbf{u}_t^{\top}\right], \quad \mathbb{E}\|\mathbf{u}_t\|_2^2 \leq \kappa_2, \quad \text{and} \quad \mathbb{E}\left[\|\mathbf{u}_t\|_2^2 \, \mathbf{u}_t \mathbf{u}_t^{\top}\right] \preceq \kappa_3 \mathbb{E}\left[\mathbf{u}_t \mathbf{u}_t^{\top}\right].$$

Consider a Gaussian noise that $\mathbf{u}_t \sim \mathcal{N}(0, \mathbf{I})$, $\forall t \in [T]$, we have $\kappa_1 = 1$, $\kappa_2 = d$, and $\kappa_3 = 3d$.

With the above assumptions, we provide some supporting lemmas below. First, we show $\varepsilon$-sensitivity of the location family given in (7).

**Lemma 1 ($\varepsilon$-Sensitivity of $\mathcal{D}(\boldsymbol{\theta})$).** *Define $\sigma_{\max}(\mathbf{A}) := \max_{\|\boldsymbol{\theta}\|_2=1}\|\mathbf{A}\boldsymbol{\theta}\|_2$. The location family given in (7) is $\varepsilon$-sensitive with parameter $\varepsilon \leq \sigma_{\max}(\mathbf{A})$. That is, for any $\boldsymbol{\theta}, \boldsymbol{\theta}' \in \boldsymbol{\Theta}$, we have $\mathcal{W}_1\left(\mathcal{D}(\boldsymbol{\theta}), \mathcal{D}\left(\boldsymbol{\theta}'\right)\right) \leq \varepsilon \left\|\boldsymbol{\theta} - \boldsymbol{\theta}'\right\|_2$, where $\mathcal{W}_1\left(\mathcal{D}, \mathcal{D}'\right)$ denotes the Wasserstein-1 distance.*

See § A of the supplementary file for the proof. Building upon Lemma 1, we have the following Lemma 2 about the performative risk $\mathrm{PR}(\boldsymbol{\theta})$.

**Lemma 2 (Lipschitz Continuity and Convexity of $\mathrm{PR}(\boldsymbol{\theta})$).** *Consider the location family given in (7). With Assumption 1 and Lemma 1, we have that: 1) the performative risk $\mathrm{PR}(\boldsymbol{\theta})$ is $L$-Lipschitz continuous for $L \leq L_{\boldsymbol{\theta}} + L_Z \sigma_{\max}(\mathbf{A})$; 2) the performative risk $\mathrm{PR}(\boldsymbol{\theta})$ is $\gamma$-strongly convex for*

$$\gamma \geq \max\left\{\gamma_{\boldsymbol{\theta}} - \beta^2/\gamma_Z, \gamma_{\boldsymbol{\theta}} - 2\varepsilon\beta + \gamma_Z \sigma_{\min}^2(\mathbf{A})\right\},$$

*where $\sigma_{\min}(\mathbf{A}) := \min_{\|\boldsymbol{\theta}\|_2=1}\|\mathbf{A}\boldsymbol{\theta}\|_2$.*

See § B of the supplementary file for the proof. Based on the Lipschitz continuity and convexity of $\mathrm{PR}(\boldsymbol{\theta})$, we provide the convergence result of the robust primal-dual framework below.

**Lemma 3 (Convergence Result of Robust Primal-Dual Framework).** *Set $\eta = \frac{1}{\sqrt{T}}$. Then, there exists a constant $\delta \in \left[\frac{1-\sqrt{1-32\eta^2 L_{\mathbf{g}}^2}}{4\eta^2}, \frac{1+\sqrt{1-32\eta^2 L_{\mathbf{g}}^2}}{4\eta^2}\right]$ such that under Assumptions 1-3, for $T \geq 32 L_{\mathbf{g}}^2$, the regret satisfies:*

$$\sum_{t=1}^{T}\left(\mathbb{E}[\mathrm{PR}(\boldsymbol{\theta}_t)] - \mathrm{PR}\left(\boldsymbol{\theta}_{\mathrm{PO}}\right)\right) \leq \frac{\gamma\sqrt{T}}{\gamma-a}\left(2R^2 + C^2 + 2L^2\right)$$
$$+ \frac{\gamma}{\gamma-a}\left(\frac{1}{2a} + \frac{1}{\sqrt{T}}\right)\sum_{t=1}^{T}\mathbb{E}\left\|\nabla_{\boldsymbol{\theta}}\widehat{\mathrm{PR}}_t(\boldsymbol{\theta}_t) - \nabla_{\boldsymbol{\theta}}\mathrm{PR}(\boldsymbol{\theta}_t)\right\|_2^2,$$

*where $a \in (0, \gamma)$ is a constant. Further, for any $i \in [m]$, the constraint violation satisfies:*

$$\mathbb{E}\left[\sum_{t=1}^{T} g_i\left(\boldsymbol{\theta}_t\right)\right] \leq \sqrt{1+\delta}\left(2R + \sqrt{2}C + 2L\right)\sqrt{T}$$
$$+ \sqrt{1+\delta}\left(\frac{T^{\frac{1}{4}}}{\sqrt{a}} + \sqrt{2}\right)\left(\sum_{t=1}^{T}\mathbb{E}\left\|\nabla_{\boldsymbol{\theta}}\widehat{\mathrm{PR}}_t(\boldsymbol{\theta}_t) - \nabla_{\boldsymbol{\theta}}\mathrm{PR}(\boldsymbol{\theta}_t)\right\|_2^2\right)^{\frac{1}{2}}.$$

**Remark 2.** *Lemma 3 reveals the impact of gradient approximation error on the convergence performance of the robust primal-dual framework. By Lemma 3, if the accumulated gradient approximation error is less than $\mathcal{O}(\sqrt{T})$, both the regret and the constraint violations are bounded by $\mathcal{O}(\sqrt{T})$. Although stochastic primal-dual methods for constrained problems without performativity also use approximated (stochastic) gradients, they generally require unbiased gradient approximation [Tan et al., 2018; Yan et al., 2019; Cao and Başar, 2022]. This requirement, however, is difficult to satisfy in performative prediction since the unknown performative effect of decisions changes the data distribution. In contrast, the robust primal-dual framework does not restrict the approximate gradients to be unbiased and hence offers more flexibility to the design of gradient approximation.*

Proof of Lemma 3 is provided in § C of the supplementary file. In the next lemma, we bound the gradient approximation error of the adaptive primal-dual algorithm.

**Lemma 4** (**Gradient Approximation Error**). *Set* $\zeta_t = \frac{2}{\kappa_1(t-1)+2\kappa_3}$, $\forall t \in [T]$. *Then, under Assumptions 4 and 5, the accumulated gradient approximation error is upper bounded by:*

$$\sum_{t=1}^T \mathbb{E}\left\|\nabla_{\boldsymbol{\theta}}\widehat{\mathrm{PR}}_t(\boldsymbol{\theta}_t) - \nabla_{\boldsymbol{\theta}}\mathrm{PR}(\boldsymbol{\theta}_t)\right\|_2^2 \leq \frac{2T\sigma^2}{n} + \frac{4}{n}\left(2L_Z^2 + \beta^2 R^2\left(1 + 2\sigma_{\max}(\mathbf{A})\right)\right)\overline{\alpha}\ln(T),$$

*where* $\overline{\alpha} := \max\left\{\frac{2\kappa_3}{\kappa_1}\|\widehat{\mathbf{A}}_0 - \mathbf{A}\|_{\mathrm{F}}^2, \frac{8\kappa_2\,\mathrm{tr}(\boldsymbol{\Sigma})}{\kappa_1^2}\right\}$. *In Algorithm 1, we set* $\widehat{\mathbf{A}}_0 = \mathbf{0}$, *and thus we have* $\overline{\alpha} = \max\left\{\frac{2\kappa_3}{\kappa_1}\|\mathbf{A}\|_{\mathrm{F}}^2, \frac{8\kappa_2\,\mathrm{tr}(\boldsymbol{\Sigma})}{\kappa_1^2}\right\}$.

**Remark 3.** *Lemma 4 demonstrates that the gradient approximation error of the adaptive primal-dual algorithm is upper bounded by* $\mathcal{O}(T/n + \ln(T))$. *If we set the number of initial samples* $n \geq \sqrt{T}$, *we have* $\sum_{t=1}^T \mathbb{E}\left\|\nabla_{\boldsymbol{\theta}}\widehat{\mathrm{PR}}(\boldsymbol{\theta}_t) - \nabla_{\boldsymbol{\theta}}\widehat{\mathrm{PR}}_t(\boldsymbol{\theta}_t)\right\|_2^2 \leq \mathcal{O}(\sqrt{T})$. *According to Lemma 3, this suffices to make the regret and constraint violation bounds to be* $\mathcal{O}(\sqrt{T})$.

Proof of Lemma 4 is presented in § D of the supplementary file. Combining Lemma 3 and Lemma 4 yields the regret and constraint violations of Algorithm 1, which is elaborated in Theorem 1 below.

---

**Theorem 1.** *Set* $\eta = \frac{1}{\sqrt{T}}$ *and* $\zeta_t = \frac{2}{\kappa_1(t-1)+2\kappa_3}$, $\forall t \in [T]$. *Then, there exists a constant* $\delta \in \left[\frac{1-\sqrt{1-32\eta^2 L_{\mathbf{g}}^2}}{4\eta^2}, \frac{1+\sqrt{1-32\eta^2 L_{\mathbf{g}}^2}}{4\eta^2}\right]$ *such that under Assumptions 1-5, for* $T \geq 32L_{\mathbf{g}}^2$, *the regret of Algorithm 1 is upper bounded by:*

$$\sum_{t=1}^T \left(\mathbb{E}[\mathrm{PR}(\boldsymbol{\theta}_t)] - \mathrm{PR}(\boldsymbol{\theta}_{\mathrm{PO}})\right) \leq \frac{\gamma\sqrt{T}}{\gamma - a}\left(2R^2 + C^2 + 2L^2\right) + \frac{\gamma\sigma^2}{\gamma - a}\left(\frac{1}{a} + \frac{2}{\sqrt{T}}\right)\frac{T}{n}$$
$$+ \frac{\gamma\overline{\alpha}\ln(T)}{n(\gamma - a)}\left(\frac{2}{a} + \frac{4}{\sqrt{T}}\right)\left(2L_Z^2 + \beta^2 R^2\left(1 + 2\sigma_{\max}(\mathbf{A})\right)\right),$$

*Further, for any* $i \in [m]$, *the constraint violation is upper bounded by:*

$$\mathbb{E}\left[\sum_{t=1}^T g_i(\boldsymbol{\theta}_t)\right] \leq \sqrt{1+\delta}\left[\left(2R + \sqrt{2}C + 2L\right)\sqrt{T} + \left(\frac{\sqrt{2}}{\sqrt{a}} + \frac{2}{T^{\frac{1}{4}}}\right)\frac{\sigma T^{\frac{3}{4}}}{\sqrt{n}}\right]$$
$$+ \frac{2\sqrt{\overline{\alpha}(1+\delta)\ln(T)}}{\sqrt{n}}\left(\frac{T^{\frac{1}{4}}}{\sqrt{a}} + \sqrt{2}\right)\left(2L_Z^2 + \beta^2 R^2\left(1 + 2\sigma_{\max}(\mathbf{A})\right)\right)^{\frac{1}{2}}.$$

---

**Remark 4.** *Theorem 1 demonstrates that Algorithm 1 achieves* $\mathcal{O}(\sqrt{T} + T/n)$ *regret and* $\mathcal{O}(\sqrt{T} + T^{\frac{3}{4}}/\sqrt{n})$ *constraint violations. By setting* $n = \sqrt{T}$, *we have* $T/n = \sqrt{T}$ *and* $T^{\frac{3}{4}}/\sqrt{n} = \sqrt{T}$, *and hence both the regret and constraint violations are upper bounded by* $\mathcal{O}(\sqrt{T})$. *This indicates that Algorithm 1 attains the same order of performance as the stochastic primal-dual algorithm without performativity [Tan et al., 2018; Yan et al., 2019].*

**Remark 5.** *Throughout the time horizon* $T$, *Algorithm 1 requires a total of* $\sqrt{T} + 2T$ *samples. Among them,* $\sqrt{T}$ *samples are dedicated to approximate the expectation over the base component* $Z_0$. *Furthermore, each iteration requires an additional 2 samples to construct the online least-square objective, accumulating the remaining* $2T$ *samples.*

## 5 Numerical Experiments

This section verifies the efficacy of our algorithm and theoretical results by conducting numerical experiments on two examples: multi-task linear regression and multi-asset portfolio.

We first consider a multi-task linear regression problem in an undirected graph $\mathcal{G} := (\mathcal{V}, \mathcal{E})$, where $\mathcal{V}$ represents the node set and $\mathcal{E}$ represents the edge set. Each node $i$ handles a linear regression task $\mathrm{PR}_i(\boldsymbol{\theta}_i) := \mathbb{E}_{(\mathbf{x}_i, y_i) \sim \mathcal{D}_i(\boldsymbol{\theta}_i)}\ell_i(\boldsymbol{\theta}_i; (\mathbf{x}_i, y_i))$, where $\boldsymbol{\theta}_i$ is the parameter vector and $(\mathbf{x}_i, y_i)$ is a feature-label pair. The loss function of each task is $\ell_i(\boldsymbol{\theta}_i; (\mathbf{x}_i, y_i)) = \frac{1}{2}(y_i - \boldsymbol{\theta}_i^\top \mathbf{x}_i)^2, \forall i \in \mathcal{V}$. The parameters of each connected node pair are subject to a proximity constraint $\|\boldsymbol{\theta}_i - \boldsymbol{\theta}_j\|_2^2 \leq b_{ij}^2$, $\forall (i, j) \in \mathcal{E}$. The entire network aims to solve the following problem:

$$\min_{\boldsymbol{\theta}_i, \forall i} \quad \frac{1}{2}\sum_{i \in \mathcal{V}}\mathbb{E}_{(\mathbf{x}_i, y_i) \sim \mathcal{D}_i(\boldsymbol{\theta}_i)}(y_i - \boldsymbol{\theta}_i^\top \mathbf{x}_i)^2 \quad \text{s.t.} \quad \frac{1}{2}\|\boldsymbol{\theta}_i - \boldsymbol{\theta}_j\|_2^2 \leq b_{ij}^2, \forall (i, j) \in \mathcal{E}.$$

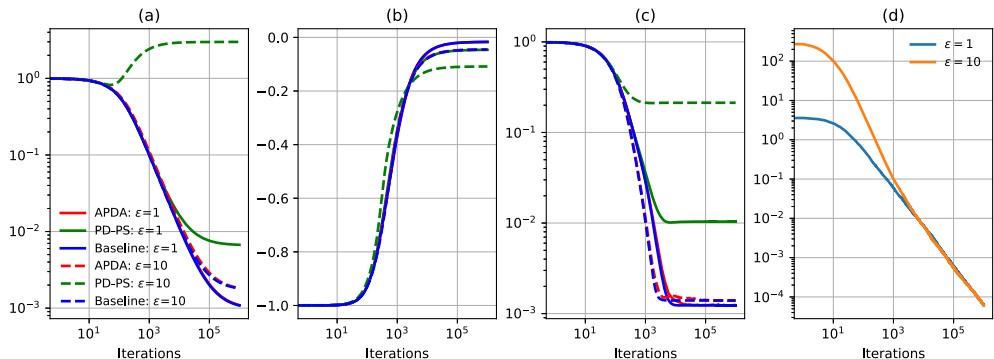

Figure 1: Multi-task linear regression. (a) $\frac{\text{Reg}(t)}{t \cdot \text{Reg}(1)}$; (b) $\frac{\text{Vio}_i(t)}{t \cdot |\text{Vio}_i(1)|}$; (c) $\|\boldsymbol{\theta}_t - \boldsymbol{\theta}_{\text{PO}}\|_2^2$; (d) $\|\widehat{\mathbf{A}}_t - \mathbf{A}\|_{\text{F}}^2$.

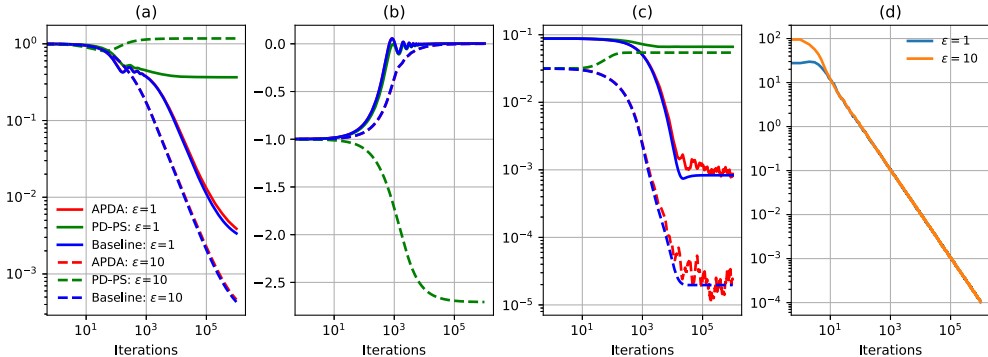

Figure 2: Multi-asset portfolio. (a) $\frac{\text{Reg}(t)}{t \cdot \text{Reg}(1)}$; (b) $\frac{\text{Vio}_i(t)}{t \cdot |\text{Vio}_i(1)|}$; (c) $\|\boldsymbol{\theta}_t - \boldsymbol{\theta}_{\text{PO}}\|_2^2$; (d) $\|\widehat{\mathbf{A}}_t - \mathbf{A}\|_{\text{F}}^2$.

The second example considers the multi-asset portfolio described in Example 1. The simulation details are provided in § F of the supplementary file.

We compare the proposed adaptive primal-dual algorithm (abbreviated as APDA) with two approaches. The first approach is "PD-PS", which stands for the primal-dual (PD) algorithm used to find the performative stable (PS) points. The algorithm PD-PS is similar to APDA, but it uses only the first term in Eq. (8) as the approximate gradient. The second approach is "baseline", which runs the same procedures as APDA with perfect knowledge of $\mathbf{A}$, i.e., the performative effect is known. We consider four performance metrics: (a) relative time-average regret $\frac{\text{Reg}(t)}{t \cdot \text{Reg}(1)}$, (b) relative time-average constraint violation $\frac{\text{Vio}_i(t)}{t \cdot |\text{Vio}_i(1)|}$, (c) decision deviation $\|\boldsymbol{\theta}_t - \boldsymbol{\theta}_{\text{PO}}\|_2^2$, and (d) parameter estimation error $\|\widehat{\mathbf{A}}_t - \mathbf{A}\|_{\text{F}}^2$.

Fig. 1 and Fig. 2 show the numerical results of the multi-task linear regression and the multi-asset portfolio, respectively. In both figures, we consider two settings for the sensitivity parameter of $\mathcal{D}(\boldsymbol{\theta})$, namely $\varepsilon = 1$ and $\varepsilon = 10$. The results of these two figures are qualitatively analogous. First, we observe that APDA outperforms PD-PS significantly that both the relative time-average regret and the decision derivation of the former achieve an accuracy around or up to $10^{-3}$ for the setting of $T = 10^6$, while these of the latter have worse performance for $\varepsilon = 1$ and converge to constants for $\varepsilon = 10$. The relative time-average constraint of all cases converges to zero or negative numbers. This corroborates the sublinearity of the regret and the constraint violations of APDA, as shown in Theorem 1. More importantly, this result implies that the larger the sensitivity parameter $\varepsilon$, the stronger the performative power is and, consequently, the worse PD-PS performs. In contrast, by tracking the performative gradient, APDA adapts to the unknown performative effect and performs well constantly. Moreover, both subfigures (d) show that the error of parameter estimates decreases sublinearly with iterations, validating the effectiveness of the online parameter estimation. Last but not least, the performance of APDA is close to the performance of the baseline, which manifests the effectiveness of our proposed APDA algorithm.

# 6 Conclusions

This paper has studied the performative prediction problem under inequality constraints, where the agnostic performative effect of decisions changes future data distributions. To find the performative optima for the problem, we have developed a robust primal-dual framework that admits inexact gradients up to an accuracy of $\mathcal{O}(\sqrt{T})$, yet delivers the same $\mathcal{O}(\sqrt{T})$ regret and constraint violations as the stochastic primal-dual algorithm without performativity. Then, based on this framework, we have proposed an adaptive primal-dual algorithm for location family with effective gradient approximation method that meets the desired accuracy using only $\sqrt{T} + 2T$ samples. Numerical experiments have validated the effectiveness of our algorithm and theoretical results.

## Acknowledgments and Disclosure of Funding

The work was supported by the National Natural Science Foundation of China Grant 62203373.

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
