## A  Proof of Lemma 1

The $\varepsilon$-sensitivity of distributions is defined below.

**Definition A1** ($\varepsilon$-**Sensitivity**). *A distribution $\mathcal{D}(\cdot)$ is called $\varepsilon$-sensitive if for any $\boldsymbol{\theta}, \boldsymbol{\theta}' \in \boldsymbol{\Theta}$ there exists a constant $\varepsilon > 0$ such that*

$$\mathcal{W}_1 \left( \mathcal{D}(\boldsymbol{\theta}), \mathcal{D}\left(\boldsymbol{\theta}'\right) \right) \le \varepsilon \left\| \boldsymbol{\theta} - \boldsymbol{\theta}' \right\|_2,$$

*where $\mathcal{W}_1 \left( \mathcal{D}, \mathcal{D}' \right)$ denotes the Wasserstein-1 distance.*

Next, we provide the following lemma.

**Lemma A1.** *Suppose that the distribution map $\mathcal{D}(\boldsymbol{\theta})$ forms a location family (7). Then, we have*

$$\mathcal{W}_1 \left( \mathcal{D}(\boldsymbol{\theta}), \mathcal{D}\left(\boldsymbol{\theta}'\right) \right) \le \left\| \mathbf{A} \left( \boldsymbol{\theta} - \boldsymbol{\theta}' \right) \right\|_2.$$

*Proof.* By definition, $\mathcal{W}_1 \left( \mathcal{D}(\boldsymbol{\theta}), \mathcal{D}\left(\boldsymbol{\theta}'\right) \right) := \inf_{\Gamma(\mathcal{D}(\boldsymbol{\theta}), \mathcal{D}(\boldsymbol{\theta}'))} \mathbb{E}_{(Z_{\boldsymbol{\theta}}, Z_{\boldsymbol{\theta}'}) \sim (\mathcal{D}(\boldsymbol{\theta}), \mathcal{D}(\boldsymbol{\theta}'))} \left\| Z_{\boldsymbol{\theta}} - Z_{\boldsymbol{\theta}'} \right\|_2$, where $\Gamma \left( \mathcal{D}(\boldsymbol{\theta}), \mathcal{D}\left(\boldsymbol{\theta}'\right) \right)$ is the set of all couplings of the distributions $\mathcal{D}(\boldsymbol{\theta})$ and $\mathcal{D}\left(\boldsymbol{\theta}'\right)$. One way to couple $\mathcal{D}(\boldsymbol{\theta})$ and $\mathcal{D}\left(\boldsymbol{\theta}'\right)$ is to set $Z_{\boldsymbol{\theta}} \sim \mathcal{D}(\boldsymbol{\theta})$ and $Z_{\boldsymbol{\theta}'} \sim \mathcal{D}\left(\boldsymbol{\theta}'\right)$. Under this setting, with the definition of $\mathcal{D}(\boldsymbol{\theta})$ (7), we have $\mathbb{E}_{(Z_{\boldsymbol{\theta}}, Z_{\boldsymbol{\theta}'}) \sim (\mathcal{D}(\boldsymbol{\theta}), \mathcal{D}(\boldsymbol{\theta}'))} \left\| Z_{\boldsymbol{\theta}} - Z_{\boldsymbol{\theta}'} \right\|_2 = \left\| \mathbf{A} \left( \boldsymbol{\theta} - \boldsymbol{\theta}' \right) \right\|_2$, and hence $\mathcal{W}_1 \left( \mathcal{D}(\boldsymbol{\theta}), \mathcal{D}\left(\boldsymbol{\theta}'\right) \right) \le \left\| \mathbf{A} \left( \boldsymbol{\theta} - \boldsymbol{\theta}' \right) \right\|_2$. $\square$

Define $\sigma_{\max}(\mathbf{A}) := \max_{\|\boldsymbol{\theta}\|_2 = 1} \|\mathbf{A}\boldsymbol{\theta}\|_2$, we have $\left\| \mathbf{A} \left( \boldsymbol{\theta} - \boldsymbol{\theta}' \right) \right\|_2 \le \sigma_{\max}(\mathbf{A}) \left\| \boldsymbol{\theta} - \boldsymbol{\theta}' \right\|_2$. By Lemma A1, $\mathcal{W}_1 \left( \mathcal{D}(\boldsymbol{\theta}), \mathcal{D}\left(\boldsymbol{\theta}'\right) \right) \le \sigma_{\max}(\mathbf{A}) \left\| \boldsymbol{\theta} - \boldsymbol{\theta}' \right\|_2$. By Definition A1, the sensitivity parameter $\varepsilon \le \sigma_{\max}(\mathbf{A})$, which proves Lemma 1.

## B  Proof of Lemma 2

*Proof of the $L$-Lipschitz continuity of* $\mathrm{PR}(\boldsymbol{\theta})$. To show the $L$-Lipschitz continuity of $\mathrm{PR}(\boldsymbol{\theta})$, it suffices to show that there exists a positive constant $L$ such that, for any $\boldsymbol{\theta}, \boldsymbol{\theta}' \in \boldsymbol{\Theta}$, $\|\mathrm{PR}(\boldsymbol{\theta}) - \mathrm{PR}(\boldsymbol{\theta}')\|_2 \le L\|\boldsymbol{\theta} - \boldsymbol{\theta}'\|_2$. By Assumption 1, the loss function $\ell(\boldsymbol{\theta}; Z)$ is $L_{\boldsymbol{\theta}}$-Lipschitz continous in $\boldsymbol{\theta}$ and $L_Z$-Lipschitz continous in $Z$, i.e.,

$$\left\| \ell\left(\boldsymbol{\theta}; Z\right) - \ell\left(\boldsymbol{\theta}'; Z'\right) \right\|_2 \le L_{\boldsymbol{\theta}} \left\| \boldsymbol{\theta} - \boldsymbol{\theta}' \right\|_2 + L_Z \left\| Z - Z' \right\|_2, \forall \boldsymbol{\theta}, \boldsymbol{\theta}' \in \boldsymbol{\Theta}, Z, Z' \in \mathbb{R}^k.$$

Then, we have

$$\left\| \mathop{\mathbb{E}}_{Z_0 \sim \mathcal{D}_0} \ell\left(\boldsymbol{\theta}; Z_0 + \mathbf{A}\boldsymbol{\theta}\right) - \mathop{\mathbb{E}}_{Z_0 \sim \mathcal{D}_0} \ell\left(\boldsymbol{\theta}'; Z_0 + \mathbf{A}\boldsymbol{\theta}'\right) \right\|_2$$
$$\le \mathop{\mathbb{E}}_{Z_0 \sim \mathcal{D}_0} \left\| \ell\left(\boldsymbol{\theta}; Z_0 + \mathbf{A}\boldsymbol{\theta}\right) - \ell\left(\boldsymbol{\theta}'; Z_0 + \mathbf{A}\boldsymbol{\theta}'\right) \right\|_2$$
$$\le L_{\boldsymbol{\theta}} \left\| \boldsymbol{\theta} - \boldsymbol{\theta}' \right\|_2 + L_Z \left\| \mathbf{A} \left( \boldsymbol{\theta} - \boldsymbol{\theta}' \right) \right\|_2$$
$$\le \left( L_{\boldsymbol{\theta}} + L_Z \sigma_{\max}(\mathbf{A}) \right) \left\| \boldsymbol{\theta} - \boldsymbol{\theta}' \right\|_2.$$

Thus, there exists a constant $L \le L_{\boldsymbol{\theta}} + L_Z \sigma_{\max}(\mathbf{A})$ such that $\left\| \mathrm{PR}(\boldsymbol{\theta}) - \mathrm{PR}(\boldsymbol{\theta}') \right\|_2 \le L \left\| \boldsymbol{\theta} - \boldsymbol{\theta}' \right\|_2$, $\forall \boldsymbol{\theta}, \boldsymbol{\theta}' \in \boldsymbol{\Theta}$, which proves the $L$-Lipschitz continuity of $\mathrm{PR}(\boldsymbol{\theta})$. $\square$

*Proof of the $\gamma$-strongly convexity of* $\mathrm{PR}(\boldsymbol{\theta})$. By Assumption 1, $\ell(\boldsymbol{\theta}; Z)$ is $\gamma_Z$-strongly convex in $Z$. Then, we have

$$\mathop{\mathbb{E}}_{Z \sim \mathcal{D}(\boldsymbol{\theta})} \ell(\boldsymbol{\theta}; Z) \ge \mathop{\mathbb{E}}_{Z \sim \mathcal{D}(\alpha\boldsymbol{\theta} + (1-\alpha)\boldsymbol{\theta}')} \ell(\boldsymbol{\theta}; Z) + \frac{(1-\alpha)^2 \gamma_Z}{2} \left\| \mathbf{A} \left( \boldsymbol{\theta} - \boldsymbol{\theta}' \right) \right\|_2^2$$
$$+ (1-\alpha) \left( \nabla_Z \mathop{\mathbb{E}}_{Z \sim \mathcal{D}(\alpha\boldsymbol{\theta} + (1-\alpha)\boldsymbol{\theta}')} \ell(\boldsymbol{\theta}; Z) \right)^{\top} \mathbf{A} \left( \boldsymbol{\theta} - \boldsymbol{\theta}' \right), \tag{b1}$$

$$\mathop{\mathbb{E}}_{Z \sim \mathcal{D}(\boldsymbol{\theta}')} \ell(\boldsymbol{\theta}; Z) \ge \mathop{\mathbb{E}}_{Z \sim \mathcal{D}(\alpha\boldsymbol{\theta} + (1-\alpha)\boldsymbol{\theta}')} \ell(\boldsymbol{\theta}; Z) + \frac{\alpha^2 \gamma_Z}{2} \left\| \mathbf{A} \left( \boldsymbol{\theta} - \boldsymbol{\theta}' \right) \right\|_2^2$$
$$- \alpha \left( \nabla_Z \mathop{\mathbb{E}}_{Z \sim \mathcal{D}(\alpha\boldsymbol{\theta} + (1-\alpha)\boldsymbol{\theta}')} \ell(\boldsymbol{\theta}; Z) \right)^{\top} \mathbf{A} \left( \boldsymbol{\theta} - \boldsymbol{\theta}' \right). \tag{b2}$$

Combining $\alpha$(b1)$+(1-\alpha)$(b2), we obtain

$$\alpha \mathop{\mathbb{E}}_{Z\sim\mathcal{D}(\boldsymbol{\theta})} \ell\left(\boldsymbol{\theta};Z\right) + (1-\alpha)\mathop{\mathbb{E}}_{Z\sim\mathcal{D}(\boldsymbol{\theta}')} \ell\left(\boldsymbol{\theta};Z\right)$$

$$\geq \mathop{\mathbb{E}}_{Z\sim\mathcal{D}(\alpha\boldsymbol{\theta}+(1-\alpha)\boldsymbol{\theta}')} \ell\left(\boldsymbol{\theta};Z\right) + \frac{\alpha(1-\alpha)\gamma_Z}{2} \left\| \mathbf{A}\left(\boldsymbol{\theta}-\boldsymbol{\theta}'\right) \right\|_2^2. \tag{b3}$$

In (b3), fixing the first augment of $\ell\left(\boldsymbol{\theta};Z\right)$ at $\boldsymbol{\theta}_0$, $\forall \boldsymbol{\theta}_0 \in \boldsymbol{\Theta}$, and substracting $\frac{\gamma_Z}{2}\left\| \mathbf{A}\left(\alpha\boldsymbol{\theta}+(1-\alpha)\boldsymbol{\theta}'\right) \right\|_2^2$ on both sides, we obtain

$$\mathop{\mathbb{E}}_{Z\sim\mathcal{D}(\alpha\boldsymbol{\theta}+(1-\alpha)\boldsymbol{\theta}')} \ell\left(\boldsymbol{\theta}_0;Z\right) - \frac{\gamma_Z}{2}\left\| \mathbf{A}\left(\alpha\boldsymbol{\theta}+(1-\alpha)\boldsymbol{\theta}'\right) \right\|_2^2$$

$$\leq \alpha \mathop{\mathbb{E}}_{Z\sim\mathcal{D}(\boldsymbol{\theta})} \ell\left(\boldsymbol{\theta}_0;Z\right) + (1-\alpha)\mathop{\mathbb{E}}_{Z\sim\mathcal{D}(\boldsymbol{\theta}')} \ell\left(\boldsymbol{\theta}_0;Z\right) - \frac{\alpha(1-\alpha)\gamma_Z}{2} \left\| \mathbf{A}\left(\boldsymbol{\theta}-\boldsymbol{\theta}'\right) \right\|_2^2$$

$$- \frac{\gamma_Z}{2} \left\| \mathbf{A}\left(\alpha\boldsymbol{\theta}+(1-\alpha)\boldsymbol{\theta}'\right) \right\|_2^2$$

$$= \alpha \left( \mathop{\mathbb{E}}_{Z\sim\mathcal{D}(\boldsymbol{\theta})} \ell\left(\boldsymbol{\theta}_0;Z\right) - \frac{\gamma_Z}{2}\left\| \mathbf{A}\boldsymbol{\theta} \right\|_2^2 \right) + (1-\alpha)\left( \mathop{\mathbb{E}}_{Z\sim\mathcal{D}(\boldsymbol{\theta}')} \ell\left(\boldsymbol{\theta}_0;Z\right) - \frac{\gamma_Z}{2}\left\| \mathbf{A}\boldsymbol{\theta}' \right\|_2^2 \right). \tag{b4}$$

Eq. (b4) demonstrates that the function $\mathbb{E}_{Z\sim\mathcal{D}(\boldsymbol{\theta})}\ell\left(\boldsymbol{\theta}_0;Z\right) - \frac{\gamma_Z}{2}\left\| \mathbf{A}\boldsymbol{\theta} \right\|_2^2$ is convex in $\boldsymbol{\theta}$ for any given $\boldsymbol{\theta}_0 \in \boldsymbol{\Theta}$. By the equivalent first-order characterization, we have

$$\mathop{\mathbb{E}}_{Z\sim\mathcal{D}(\boldsymbol{\theta}')}\ell\left(\boldsymbol{\theta}_0;Z\right) \geq \frac{\gamma_Z}{2}\left\| \mathbf{A}\boldsymbol{\theta}' \right\|_2^2 + \mathop{\mathbb{E}}_{Z\sim\mathcal{D}(\boldsymbol{\theta})}\ell\left(\boldsymbol{\theta}_0;Z\right) - \frac{\gamma_Z}{2}\left\| \mathbf{A}\boldsymbol{\theta} \right\|_2^2$$

$$+ \left( \mathop{\mathbb{E}}_{Z\sim\mathcal{D}(\boldsymbol{\theta})} \mathbf{A}^\top \nabla_Z \ell\left(\boldsymbol{\theta}_0;Z\right) \right)^\top \left(\boldsymbol{\theta}'-\boldsymbol{\theta}\right) - \gamma_Z\left(\mathbf{A}^\top\mathbf{A}\boldsymbol{\theta}\right)^\top\left(\boldsymbol{\theta}'-\boldsymbol{\theta}\right)$$

$$= \mathop{\mathbb{E}}_{Z\sim\mathcal{D}(\boldsymbol{\theta})}\ell\left(\boldsymbol{\theta}_0;Z\right) + \left( \mathop{\mathbb{E}}_{Z\sim\mathcal{D}(\boldsymbol{\theta})} \mathbf{A}^\top \nabla_Z \ell\left(\boldsymbol{\theta}_0;Z\right) \right)^\top \left(\boldsymbol{\theta}'-\boldsymbol{\theta}\right) + \frac{\gamma_Z}{2}\left\| \mathbf{A}\left(\boldsymbol{\theta}-\boldsymbol{\theta}'\right) \right\|_2^2.$$

Setting $\boldsymbol{\theta}_0 = \boldsymbol{\theta}$ gives

$$\left( \mathop{\mathbb{E}}_{Z\sim\mathcal{D}(\boldsymbol{\theta})} \mathbf{A}^\top \nabla_Z \ell\left(\boldsymbol{\theta};Z\right) \right)^\top \left(\boldsymbol{\theta}'-\boldsymbol{\theta}\right)$$

$$\leq \mathop{\mathbb{E}}_{Z\sim\mathcal{D}(\boldsymbol{\theta}')}\ell\left(\boldsymbol{\theta};Z\right) - \mathop{\mathbb{E}}_{Z\sim\mathcal{D}(\boldsymbol{\theta})}\ell\left(\boldsymbol{\theta};Z\right) - \frac{\gamma_Z}{2}\left\| \mathbf{A}\left(\boldsymbol{\theta}-\boldsymbol{\theta}'\right) \right\|_2^2. \tag{b5}$$

Further, since $\ell(\boldsymbol{\theta};Z)$ is $\gamma_{\boldsymbol{\theta}}$-strongly convex in $\boldsymbol{\theta}$, we have

$$\mathop{\mathbb{E}}_{Z\sim\mathcal{D}(\boldsymbol{\theta}')}\ell\left(\boldsymbol{\theta};Z\right) \leq \mathop{\mathbb{E}}_{Z\sim\mathcal{D}(\boldsymbol{\theta}')}\ell\left(\boldsymbol{\theta}';Z\right) - \left( \mathop{\mathbb{E}}_{Z\sim\mathcal{D}(\boldsymbol{\theta}')} \nabla_{\boldsymbol{\theta}} \ell\left(\boldsymbol{\theta};Z\right) \right)^\top\left(\boldsymbol{\theta}'-\boldsymbol{\theta}\right) - \frac{\gamma_{\boldsymbol{\theta}}}{2}\left\| \boldsymbol{\theta}-\boldsymbol{\theta}' \right\|_2^2. \tag{b6}$$

Plugging (b6) into (b5) yields

$$\left( \mathop{\mathbb{E}}_{Z\sim\mathcal{D}(\boldsymbol{\theta})} \mathbf{A}^\top \nabla_Z \ell\left(\boldsymbol{\theta};Z\right) \right)^\top \left(\boldsymbol{\theta}'-\boldsymbol{\theta}\right) + \left( \mathop{\mathbb{E}}_{Z\sim\mathcal{D}(\boldsymbol{\theta}')} \nabla_{\boldsymbol{\theta}} \ell\left(\boldsymbol{\theta};Z\right) \right)^\top \left(\boldsymbol{\theta}'-\boldsymbol{\theta}\right)$$

$$\leq \mathop{\mathbb{E}}_{Z\sim\mathcal{D}(\boldsymbol{\theta}')}\ell\left(\boldsymbol{\theta}';Z\right) - \mathop{\mathbb{E}}_{Z\sim\mathcal{D}(\boldsymbol{\theta})}\ell\left(\boldsymbol{\theta};Z\right) - \frac{\gamma_Z}{2}\left\| \mathbf{A}\left(\boldsymbol{\theta}-\boldsymbol{\theta}'\right) \right\|_2^2 - \frac{\gamma_{\boldsymbol{\theta}}}{2}\left\| \boldsymbol{\theta}-\boldsymbol{\theta}' \right\|_2^2.$$

Rearranging the terms in the above inequality gives

$$\mathrm{PR}(\boldsymbol{\theta}') \geq \mathrm{PR}(\boldsymbol{\theta}) + \nabla_{\boldsymbol{\theta}}\mathrm{PR}(\boldsymbol{\theta})\left(\boldsymbol{\theta}'-\boldsymbol{\theta}\right) + \frac{\gamma_Z}{2}\left\| \mathbf{A}\left(\boldsymbol{\theta}-\boldsymbol{\theta}'\right) \right\|_2^2 + \frac{\gamma_{\boldsymbol{\theta}}}{2}\left\| \boldsymbol{\theta}-\boldsymbol{\theta}' \right\|_2^2$$

$$+ \left( \mathop{\mathbb{E}}_{Z\sim\mathcal{D}(\boldsymbol{\theta}')} \nabla_{\boldsymbol{\theta}} \ell\left(\boldsymbol{\theta};Z\right) - \mathop{\mathbb{E}}_{Z\sim\mathcal{D}(\boldsymbol{\theta})} \nabla_{\boldsymbol{\theta}} \ell\left(\boldsymbol{\theta};Z\right) \right)^\top \left(\boldsymbol{\theta}'-\boldsymbol{\theta}\right). \tag{b7}$$

By the $\beta$-smoothness of $\mathrm{PR}(\boldsymbol{\theta})$, we have

$$\left( \mathop{\mathbb{E}}_{Z\sim\mathcal{D}(\boldsymbol{\theta}')} \nabla_{\boldsymbol{\theta}} \ell\left(\boldsymbol{\theta};Z\right) - \mathop{\mathbb{E}}_{Z\sim\mathcal{D}(\boldsymbol{\theta})} \nabla_{\boldsymbol{\theta}} \ell\left(\boldsymbol{\theta};Z\right) \right)^\top \left(\boldsymbol{\theta}'-\boldsymbol{\theta}\right)$$

$$\geq -\beta\left\| \mathbf{A}\left(\boldsymbol{\theta}-\boldsymbol{\theta}'\right) \right\|_2 \left\| \boldsymbol{\theta}-\boldsymbol{\theta}' \right\|_2$$

$$\geq -\frac{\gamma_Z}{2}\left\| \mathbf{A}\left(\boldsymbol{\theta}-\boldsymbol{\theta}'\right) \right\|_2^2 - \frac{\beta^2}{2\gamma_Z}\left\| \boldsymbol{\theta}-\boldsymbol{\theta}' \right\|_2^2. \tag{b8}$$

Plugging (b8) into (b7) yields

$$\text{PR}(\boldsymbol{\theta}') \geq \text{PR}(\boldsymbol{\theta}) + \nabla_{\boldsymbol{\theta}}\text{PR}(\boldsymbol{\theta})\left(\boldsymbol{\theta}' - \boldsymbol{\theta}\right) + \frac{1}{2}\left(\gamma_{\boldsymbol{\theta}} - \frac{\beta^2}{\gamma_Z}\right)\left\|\boldsymbol{\theta} - \boldsymbol{\theta}'\right\|_2^2.$$

Therefore, the convexity parameter of $\text{PR}(\boldsymbol{\theta})$ satisfies $\gamma \geq \gamma_{\boldsymbol{\theta}} - \frac{\beta^2}{\gamma_Z}$. In addition, by the $\varepsilon$-sensitivity of $\text{PR}(\boldsymbol{\theta})$, we have

$$\left(\mathop{\mathbb{E}}_{Z\sim\mathcal{D}(\boldsymbol{\theta}')}\nabla_{\boldsymbol{\theta}}\ell\left(\boldsymbol{\theta}; Z\right) - \mathop{\mathbb{E}}_{Z_0\sim\mathcal{D}(\boldsymbol{\theta})}\nabla_{\boldsymbol{\theta}}\ell\left(\boldsymbol{\theta}; Z\right)\right)^{\top}\left(\boldsymbol{\theta}' - \boldsymbol{\theta}\right) \geq -\varepsilon\beta\left\|\boldsymbol{\theta} - \boldsymbol{\theta}'\right\|_2^2. \tag{b9}$$

Plugging (b9) into (b7) yields

$$\text{PR}(\boldsymbol{\theta}') \geq \text{PR}(\boldsymbol{\theta}) + \nabla_{\boldsymbol{\theta}}\text{PR}(\boldsymbol{\theta})\left(\boldsymbol{\theta}' - \boldsymbol{\theta}\right) + \frac{1}{2}\left(\gamma_{\boldsymbol{\theta}} - 2\varepsilon\beta + \gamma_Z\sigma_{\min}^2(\mathbf{A})\right)\left\|\boldsymbol{\theta} - \boldsymbol{\theta}'\right\|_2^2,$$

where $\sigma_{\min}(\mathbf{A}) := \min_{\|\boldsymbol{\theta}\|_2=1}\|\mathbf{A}\boldsymbol{\theta}\|_2$. Thus, we also have $\gamma \geq \gamma_{\boldsymbol{\theta}} - 2\varepsilon\beta + \gamma_Z\sigma_{\min}^2(\mathbf{A})$. Combining the above results, we obtain $\gamma \geq \max\left\{\gamma_{\boldsymbol{\theta}} - \beta^2/\gamma_Z, \gamma_{\boldsymbol{\theta}} - 2\varepsilon\beta + \gamma_Z\sigma_{\min}^2(\mathbf{A})\right\}$, which proves the $\gamma$-strongly convexity of $\text{PR}(\boldsymbol{\theta})$. $\qquad\square$

## C   Proof of Lemma 3

The proof of Lemma 3 utilizes the following two supporting lemmas.

**Lemma C1.** *Consider the update steps* (5) *and* (6). *Under Assumptions 1-3, for any $\boldsymbol{\theta} \in \Theta$, $\boldsymbol{\lambda} \in \mathbb{R}_+^m$, and $t \in [T]$, the Lagrangian* (2) *satisfies:*

$$\sum_{t=1}^{T}\left(\mathcal{L}\left(\boldsymbol{\theta}_t, \boldsymbol{\lambda}\right) - \mathcal{L}\left(\boldsymbol{\theta}, \boldsymbol{\lambda}_t\right)\right) \leq \frac{2R^2}{\eta} + \frac{\|\boldsymbol{\lambda}\|_2^2}{2\eta} + \frac{\eta}{2}\sum_{t=1}^{T}\|\nabla_{\boldsymbol{\lambda}}\mathcal{L}(\boldsymbol{\theta}_t, \boldsymbol{\lambda}_t)\|_2^2 + \frac{\eta}{2}\sum_{t=1}^{T}\left\|\nabla_{\boldsymbol{\theta}}\widehat{\mathcal{L}}_t(\boldsymbol{\theta}_t, \boldsymbol{\lambda}_t)\right\|_2^2$$
$$+ \sum_{t=1}^{T}\left\langle\boldsymbol{\theta}_t - \boldsymbol{\theta}, \nabla_{\boldsymbol{\theta}}\mathcal{L}\left(\boldsymbol{\theta}_t, \boldsymbol{\lambda}_t\right) - \nabla_{\boldsymbol{\theta}}\widehat{\mathcal{L}}_t\left(\boldsymbol{\theta}_t, \boldsymbol{\lambda}_t\right)\right\rangle, \tag{c1}$$

*where $\mathbb{R}_+$ represents the set of non-negative real numbers.*

Lemma C1 establishes a relationship between the Lagrangian (2) and the primal and dual variables in the robust primal-dual framework. In particular, in (c1), the last term is introduced due to the gradient approximation. If the approximate gradient $\nabla_{\boldsymbol{\theta}}\widehat{\mathcal{L}}_t\left(\boldsymbol{\theta}_t, \boldsymbol{\lambda}_t\right)$ is unbiased, we have $\mathbb{E}\left[\nabla_{\boldsymbol{\theta}}\mathcal{L}\left(\boldsymbol{\theta}_t, \boldsymbol{\lambda}_t\right) - \nabla_{\boldsymbol{\theta}}\widehat{\mathcal{L}}_t\left(\boldsymbol{\theta}_t, \boldsymbol{\lambda}_t\right)\right] = \mathbf{0}$. Then, the last term in (c1) is eliminated by taking expectation. This is often the case in stochastic optimization without performativity [Tan et al., 2018; Yan et al., 2019; Cao and Başar, 2022]. However, in performative prediction, it is difficult to construct an unbiased gradient approximation because the unknown performative effect of decisions changes data distributions. Therefore, we must carry out the worst-case analysis on this term. In next lemma, we bound the $\ell_2$ norms of the gradients $\|\nabla_{\boldsymbol{\lambda}}\mathcal{L}(\boldsymbol{\theta}_t, \boldsymbol{\lambda}_t)\|_2^2$ and $\left\|\nabla_{\boldsymbol{\theta}}\widehat{\mathcal{L}}_t(\boldsymbol{\theta}_t, \boldsymbol{\lambda}_t)\right\|_2^2$ in (c1).

**Lemma C2.** *For any $t \in [T]$, the gradients $\nabla_{\boldsymbol{\lambda}}\mathcal{L}(\boldsymbol{\theta}_t, \boldsymbol{\lambda}_t)$ and $\nabla_{\boldsymbol{\theta}}\widehat{\mathcal{L}}_t(\boldsymbol{\theta}_t, \boldsymbol{\lambda}_t)$ respectively satisfy:*

1. $\|\nabla_{\boldsymbol{\lambda}}\mathcal{L}(\boldsymbol{\theta}_t, \boldsymbol{\lambda}_t)\|_2^2 \leq 2C^2 + 2\delta^2\eta^2\|\boldsymbol{\lambda}_t\|_2^2$;

2. $\left\|\nabla_{\boldsymbol{\theta}}\widehat{\mathcal{L}}_t(\boldsymbol{\theta}_t, \boldsymbol{\lambda}_t)\right\|_2^2 \leq 4L^2 + 4L_{\mathbf{g}}^2\|\boldsymbol{\lambda}_t\|_2^2 + 2\left\|\nabla_{\boldsymbol{\theta}}\widehat{\text{PR}}_t(\boldsymbol{\theta}_t) - \nabla_{\boldsymbol{\theta}}\text{PR}(\boldsymbol{\theta}_t)\right\|_2^2.$

Note that the bound of $\left\|\nabla_{\boldsymbol{\theta}}\widehat{\mathcal{L}}_t(\boldsymbol{\theta}_t, \boldsymbol{\lambda}_t)\right\|_2^2$ involves the term $\left\|\nabla_{\boldsymbol{\theta}}\widehat{\text{PR}}_t(\boldsymbol{\theta}_t) - \nabla_{\boldsymbol{\theta}}\text{PR}(\boldsymbol{\theta}_t)\right\|_2^2$, which is the gradient approximation error at the $t$th iteration. Proofs of Lemma C1 and Lemma C2 are respectively given in § C.1 and § C.1. With these two Lemmas, we are ready to prove Lemma 3.

*Proof of Lemma 3.* By Lemma C1, we have

$$\sum_{t=1}^{T} \left( \mathcal{L}\left(\boldsymbol{\theta}_t, \boldsymbol{\lambda}\right) - \mathcal{L}\left(\boldsymbol{\theta}, \boldsymbol{\lambda}_t\right) \right) \leq \frac{2R^2}{\eta} + \frac{\|\boldsymbol{\lambda}\|_2^2}{2\eta} + \frac{\eta}{2} \sum_{t=1}^{T} \|\nabla_{\boldsymbol{\lambda}} \mathcal{L}(\boldsymbol{\theta}_t, \boldsymbol{\lambda}_t)\|_2^2 + \frac{\eta}{2} \sum_{t=1}^{T} \left\| \nabla_{\boldsymbol{\theta}} \widehat{\mathcal{L}}_t(\boldsymbol{\theta}_t, \boldsymbol{\lambda}_t) \right\|_2^2$$

$$+ \frac{a}{2} \sum_{t=1}^{T} \|\boldsymbol{\theta}_t - \boldsymbol{\theta}\|_2^2 + \frac{1}{2a} \sum_{t=1}^{T} \left\| \nabla_{\boldsymbol{\theta}} \widehat{\mathrm{PR}}_t(\boldsymbol{\theta}_t) - \nabla_{\boldsymbol{\theta}} \mathrm{PR}(\boldsymbol{\theta}_t) \right\|_2^2, \quad \text{(c2)}$$

where $a > 0$ is a constant. Note that in (c2), we utilize the follwing inequality:

$$\left\langle \boldsymbol{\theta}_t - \boldsymbol{\theta}, \nabla_{\boldsymbol{\theta}} \mathcal{L}\left(\boldsymbol{\theta}_t, \boldsymbol{\lambda}_t\right) - \nabla_{\boldsymbol{\theta}} \widehat{\mathcal{L}}_t\left(\boldsymbol{\theta}_t, \boldsymbol{\lambda}_t\right) \right\rangle = \left\langle \boldsymbol{\theta}_t - \boldsymbol{\theta}, \nabla_{\boldsymbol{\theta}} \mathrm{PR}\left(\boldsymbol{\theta}_t\right) - \nabla_{\boldsymbol{\theta}} \widehat{\mathrm{PR}}_t(\boldsymbol{\theta}_t) \right\rangle$$

$$\leq \frac{a}{2} \|\boldsymbol{\theta}_t - \boldsymbol{\theta}\|_2^2 + \frac{1}{2a} \left\| \nabla_{\boldsymbol{\theta}} \mathrm{PR}(\boldsymbol{\theta}_t) - \nabla_{\boldsymbol{\theta}} \widehat{\mathrm{PR}}_t(\boldsymbol{\theta}_t) \right\|_2^2.$$

Taking expectation over (c2) and plugging into the results in Lemma C2, we have

$$\sum_{t=1}^{T} \left( \mathbb{E}[\mathrm{PR}(\boldsymbol{\theta}_t)] - \mathrm{PR}\left(\boldsymbol{\theta}_{\mathrm{PO}}\right) \right) + \sum_{t=1}^{T} \mathbb{E} \left\langle \boldsymbol{\lambda}, \mathbf{g}(\boldsymbol{\theta}_t) \right\rangle - \sum_{t=1}^{T} \mathbb{E} \left\langle \boldsymbol{\lambda}_t, \mathbf{g}(\boldsymbol{\theta}_{\mathrm{PO}}) \right\rangle - \frac{\delta \eta T}{2} \|\boldsymbol{\lambda}\|_2^2 + \frac{\delta \eta}{2} \sum_{t=1}^{T} \mathbb{E} \|\boldsymbol{\lambda}_t\|_2^2$$

$$\leq \frac{2R^2}{\eta} + \frac{\|\boldsymbol{\lambda}\|_2^2}{2\eta} + \eta T \left( C^2 + 2L^2 \right) + \eta \left( \delta^2 \eta^2 + 2L_{\mathbf{g}}^2 \right) \sum_{t=1}^{T} \mathbb{E} \|\boldsymbol{\lambda}_t\|_2^2$$

$$+ \frac{a}{2} \sum_{t=1}^{T} \mathbb{E} \|\boldsymbol{\theta}_t - \boldsymbol{\theta}\|_2^2 + \left( \frac{1}{2a} + \eta \right) \sum_{t=1}^{T} \mathbb{E} \left\| \nabla_{\boldsymbol{\theta}} \widehat{\mathrm{PR}}_t(\boldsymbol{\theta}_t) - \nabla_{\boldsymbol{\theta}} \mathrm{PR}(\boldsymbol{\theta}_t) \right\|_2^2, \quad \text{(c3)}$$

where we set $\boldsymbol{\theta}$ to $\boldsymbol{\theta}_{\mathrm{PO}}$ since any $\boldsymbol{\theta} \in \boldsymbol{\Theta}$ satisfies (c2). In (c3), the term $\sum_{t=1}^{T} \boldsymbol{\lambda}_t^\top \mathbf{g}(\boldsymbol{\theta}_{\mathrm{PO}})$ on the left side is non-positive and can be omitted, because we always have $\boldsymbol{\lambda}_t \geq \mathbf{0}$ and $\mathbf{g}(\boldsymbol{\theta}_{\mathrm{PO}}) \leq \mathbf{0}, \forall t \in [T]$. Then, rearranging the term in (c3) gives

$$\sum_{t=1}^{T} \left( \mathbb{E}[\mathrm{PR}(\boldsymbol{\theta}_t)] - \mathrm{PR}\left(\boldsymbol{\theta}_{\mathrm{PO}}\right) \right) + \sum_{t=1}^{T} \mathbb{E} \left\langle \boldsymbol{\lambda}, \mathbf{g}(\boldsymbol{\theta}_t) \right\rangle - \frac{1}{2} \left( \frac{1}{\eta} + \delta \eta T \right) \|\boldsymbol{\lambda}\|_2^2$$

$$\leq \frac{\eta}{2} \left( 2\delta^2 \eta^2 - \delta + 4L_{\mathbf{g}}^2 \right) \sum_{t=1}^{T} \mathbb{E} \|\boldsymbol{\lambda}_t\|_2^2 + \frac{2R^2}{\eta} + \eta T \left( C^2 + 2L^2 \right)$$

$$+ \frac{a}{2} \sum_{t=1}^{T} \mathbb{E} \|\boldsymbol{\theta}_t - \boldsymbol{\theta}\|_2^2 + \left( \frac{1}{2a} + \eta \right) \sum_{t=1}^{T} \mathbb{E} \left\| \nabla_{\boldsymbol{\theta}} \widehat{\mathrm{PR}}_t(\boldsymbol{\theta}_t) - \nabla_{\boldsymbol{\theta}} \mathrm{PR}(\boldsymbol{\theta}_t) \right\|_2^2. \quad \text{(c4)}$$

In (c4), the first term can be removed by properly choosing the stepsize $\eta$ and the parameter $\delta$, so that the coefficient $\frac{\eta}{2} \left( 2\delta^2 \eta^2 - \delta + 4L_{\mathbf{g}}^2 \right) \leq 0$. Since $2\delta^2 \eta^2 - \delta + 4L_{\mathbf{g}}^2$ is quadratic in $\eta$ and $\eta > 0$, the following range of $\delta$ meets the desired inequality:

$$\delta \in \left[ \frac{1 - \sqrt{1 - 32\eta^2 L_{\mathbf{g}}^2}}{4\eta^2}, \frac{1 + \sqrt{1 - 32\eta^2 L_{\mathbf{g}}^2}}{4\eta^2} \right].$$

We set $\eta = \frac{1}{\sqrt{T}}$. To guarantee that the value of $\delta$ within the above interval is a real number, we require $1 - 32\eta^2 L_{\mathbf{g}}^2 \geq 0$, i.e., the time horizon $T \geq 32L_{\mathbf{g}}^2$.

Next, we deal with the term $\frac{a}{2} \sum_{t=1}^{T} \mathbb{E} \|\boldsymbol{\theta}_t - \boldsymbol{\theta}\|_2^2$ in (c4). By the $\gamma$-convexity of the performative risk $\mathrm{PR}(\boldsymbol{\theta})$ give in Lemma 2, for any $\boldsymbol{\theta}_t \in \boldsymbol{\Theta}$, we have

$$\mathrm{PR}(\boldsymbol{\theta}_t) \geq \mathrm{PR}\left(\boldsymbol{\theta}_{\mathrm{PO}}\right) + \left\langle \nabla_{\boldsymbol{\theta}} \mathrm{PR}\left(\boldsymbol{\theta}_{\mathrm{PO}}\right), \boldsymbol{\theta}_t - \boldsymbol{\theta}_{\mathrm{PO}} \right\rangle + \frac{\gamma}{2} \|\boldsymbol{\theta}_t - \boldsymbol{\theta}_{\mathrm{PO}}\|_2^2.$$

From the optimality conditions, $\left\langle \nabla_{\boldsymbol{\theta}} \mathrm{PR}\left(\boldsymbol{\theta}_{\mathrm{PO}}\right), \boldsymbol{\theta}_t - \boldsymbol{\theta}_{\mathrm{PO}} \right\rangle \geq 0, \forall t \in [T]$. Then, we have

$$\frac{a}{2} \sum_{t=1}^{T} \mathbb{E} \|\boldsymbol{\theta}_t - \boldsymbol{\theta}_{\mathrm{PO}}\|_2^2 \leq \sum_{t=1}^{T} \frac{a}{\gamma} \left( \mathbb{E}[\mathrm{PR}(\boldsymbol{\theta}_t)] - \mathrm{PR}\left(\boldsymbol{\theta}_{\mathrm{PO}}\right) \right).$$

Further, since any $\boldsymbol{\lambda} \in \mathbb{R}_+^m$ satisfies Eq. (c4), we set $\boldsymbol{\lambda} = \frac{\left[\mathbb{E}\left[\sum_{t=1}^T \mathbf{g}(\boldsymbol{\theta}_t)\right]\right]^+}{\frac{1}{\eta} + \delta\eta T}$. With the above results, we obtain

$$\left(1 - \frac{a}{\gamma}\right) \sum_{t=1}^T \left(\mathbb{E}[\text{PR}(\boldsymbol{\theta}_t)] - \text{PR}\left(\boldsymbol{\theta}_{\text{PO}}\right)\right) + \frac{\left\|\left[\mathbb{E}\left[\sum_{t=1}^T \mathbf{g}\left(\boldsymbol{\theta}_t\right)\right]\right]^+\right\|_2^2}{2(1+\delta)\sqrt{T}}$$

$$\leq \sqrt{T}\left(2R^2 + C^2 + 2L^2\right) + \left(\frac{1}{2a} + \frac{1}{\sqrt{T}}\right) \sum_{t=1}^T \mathbb{E}\left\|\nabla_{\boldsymbol{\theta}} \widehat{\text{PR}}_t(\boldsymbol{\theta}_t) - \nabla_{\boldsymbol{\theta}} \text{PR}(\boldsymbol{\theta}_t)\right\|_2^2. \quad \text{(c5)}$$

Choosing $a \in (0, \gamma)$ and omitting the second term (non-negative) on the left side of (c5), we obtain

$$\sum_{t=1}^T \left(\mathbb{E}[\text{PR}(\boldsymbol{\theta}_t)] - \text{PR}\left(\boldsymbol{\theta}_{\text{PO}}\right)\right) \leq \frac{\gamma\sqrt{T}}{\gamma - a}\left(2R^2 + C^2 + 2L^2\right)$$

$$+ \frac{\gamma}{\gamma - a}\left(\frac{1}{2a} + \frac{1}{\sqrt{T}}\right) \sum_{t=1}^T \mathbb{E}\left\|\nabla_{\boldsymbol{\theta}} \widehat{\text{PR}}_t(\boldsymbol{\theta}_t) - \nabla_{\boldsymbol{\theta}} \text{PR}(\boldsymbol{\theta}_t)\right\|_2^2,$$

which proves the regret bound in Lemma 3. Similarly, with $a \in (0, \gamma)$, the first term on the left side of (c5) is also non-negative. Omitting it gives

$$\frac{\sum_{i=1}^m \left(\left[\mathbb{E}\left[\sum_{t=1}^T g_i\left(\boldsymbol{\theta}_t\right)\right]\right]^+\right)^2}{2(1+\delta)\sqrt{T}} \leq \sqrt{T}\left(2R^2 + C^2 + 2L^2\right)$$

$$+ \left(\frac{1}{2a} + \frac{1}{\sqrt{T}}\right) \sum_{t=1}^T \mathbb{E}\left\|\nabla_{\boldsymbol{\theta}} \widehat{\text{PR}}_t(\boldsymbol{\theta}_t) - \nabla_{\boldsymbol{\theta}} \text{PR}(\boldsymbol{\theta}_t)\right\|_2^2. \quad \text{(c6)}$$

For each $\left(\left[\mathbb{E}\left[\sum_{t=1}^T g_i\left(\boldsymbol{\theta}_t\right)\right]\right]^+\right)^2, i \in [m]$, the above inequality also holds. Then, taking the square root on both sides of (c6) and using the inequality $\sqrt{a+b+c} \leq \sqrt{a} + \sqrt{b} + \sqrt{c}, \forall a, b, c \geq 0$, we obtain

$$\left[\mathbb{E}\left[\sum_{t=1}^T g_i\left(\boldsymbol{\theta}_t\right)\right]\right]^+ \leq \sqrt{1+\delta}\left(2R + \sqrt{2}C + 2L\right)\sqrt{T}$$

$$+ \sqrt{1+\delta}\left(\frac{T^{\frac{1}{4}}}{\sqrt{a}} + \sqrt{2}\right)\left(\sum_{t=1}^T \mathbb{E}\left\|\nabla_{\boldsymbol{\theta}} \widehat{\text{PR}}_t(\boldsymbol{\theta}_t) - \nabla_{\boldsymbol{\theta}} \text{PR}(\boldsymbol{\theta}_t)\right\|_2^2\right)^{\frac{1}{2}}.$$

As $\mathbb{E}\left[\sum_{t=1}^T g_i\left(\boldsymbol{\theta}_t\right)\right] \leq \left[\mathbb{E}\left[\sum_{t=1}^T g_i\left(\boldsymbol{\theta}_t\right)\right]\right]^+$, the constraint violation result in Lemma 3 is derived. $\square$

### C.1  Proof of Lemma C1

The proof of Lemma C1 utilizes the following fact about a property of the projection operator.

**Fact C1.** *Suppose that set $\mathcal{A} \subset \mathbb{R}^d$ is closed and convex. Then, for any $\mathbf{y} \in \mathbb{R}^d$ and $\mathbf{x} \in \mathcal{A}$, we have*

$$\|\mathbf{x} - \Pi_{\mathcal{A}}(\mathbf{y})\|_2 \leq \|\mathbf{x} - \mathbf{y}\|_2,$$

*where $\Pi_{\mathcal{A}}(\mathbf{y})$ denotes the projection of $\mathbf{y}$ onto the set $\mathcal{A}$.*

With Fact C1, the proof of Lemma C1 is given below.

From Lemma 2 and Assumption 3, we know that $\mathcal{L}\left(\boldsymbol{\theta}, \boldsymbol{\lambda}_t\right)$ is convex in $\boldsymbol{\theta}$. Then, we have

$$\mathcal{L}\left(\boldsymbol{\theta}, \boldsymbol{\lambda}_t\right) \geq \mathcal{L}\left(\boldsymbol{\theta}_t, \boldsymbol{\lambda}_t\right) + \left\langle \nabla_{\boldsymbol{\theta}} \mathcal{L}\left(\boldsymbol{\theta}_t, \boldsymbol{\lambda}_t\right), \boldsymbol{\theta} - \boldsymbol{\theta}_t \right\rangle.$$

Similarly, since $\mathcal{L}(\mathbf{x}, \boldsymbol{\lambda})$ is concave in $\boldsymbol{\lambda}$, we have

$$\mathcal{L}(\boldsymbol{\theta}_t, \boldsymbol{\lambda}) \leq \mathcal{L}(\boldsymbol{\theta}_t, \boldsymbol{\lambda}_t) + \langle \nabla_{\boldsymbol{\lambda}} \mathcal{L}(\boldsymbol{\theta}_t, \boldsymbol{\lambda}_t), \boldsymbol{\lambda} - \boldsymbol{\lambda}_t \rangle.$$

Combining the above two inequalities yields

$$\mathcal{L}(\boldsymbol{\theta}_t, \boldsymbol{\lambda}) - \mathcal{L}(\boldsymbol{\theta}, \boldsymbol{\lambda}_t) \leq \langle \nabla_{\boldsymbol{\lambda}} \mathcal{L}(\boldsymbol{\theta}_t, \boldsymbol{\lambda}_t), \boldsymbol{\lambda} - \boldsymbol{\lambda}_t \rangle - \langle \nabla_{\boldsymbol{\theta}} \mathcal{L}(\boldsymbol{\theta}_t, \boldsymbol{\lambda}_t), \boldsymbol{\theta} - \boldsymbol{\theta}_t \rangle. \tag{c7}$$

From the update rule of $\boldsymbol{\lambda}$ given in (6), we have

$$\begin{aligned}
\|\boldsymbol{\lambda} - \boldsymbol{\lambda}_{t+1}\|_2^2 &= \left\| \boldsymbol{\lambda} - [\boldsymbol{\lambda}_t + \eta \nabla_{\boldsymbol{\lambda}} \mathcal{L}(\boldsymbol{\theta}_t, \boldsymbol{\lambda}_t)]^+ \right\|_2^2 \\
&\overset{(a)}{\leq} \|\boldsymbol{\lambda} - \boldsymbol{\lambda}_t\|_2^2 + \eta^2 \|\nabla_{\boldsymbol{\lambda}} \mathcal{L}(\boldsymbol{\theta}_t, \boldsymbol{\lambda}_t)\|_2^2 - 2\eta \langle \boldsymbol{\lambda} - \boldsymbol{\lambda}_t, \nabla_{\boldsymbol{\lambda}} \mathcal{L}(\boldsymbol{\theta}_t, \boldsymbol{\lambda}_t) \rangle,
\end{aligned} \tag{c8}$$

where $(a)$ is based on Fact C1. Rearranging the terms in (c8) gives

$$\langle \boldsymbol{\lambda} - \boldsymbol{\lambda}_t, \nabla_{\boldsymbol{\lambda}} \mathcal{L}(\boldsymbol{\theta}_t, \boldsymbol{\lambda}_t) \rangle \leq \frac{1}{2\eta} \left( \|\boldsymbol{\lambda} - \boldsymbol{\lambda}_t\|_2^2 - \|\boldsymbol{\lambda} - \boldsymbol{\lambda}_{t+1}\|_2^2 \right) + \frac{\eta}{2} \|\nabla_{\boldsymbol{\lambda}} \mathcal{L}(\boldsymbol{\theta}_t, \boldsymbol{\lambda}_t)\|_2^2. \tag{c9}$$

Similarly, from the update rule of $\boldsymbol{\theta}$ given in (5), we have

$$\begin{aligned}
\|\boldsymbol{\theta} - \boldsymbol{\theta}_{t+1}\|_2^2 &= \left\| \boldsymbol{\theta} - \Pi_{\boldsymbol{\Theta}} \left( \boldsymbol{\theta}_t - \eta \nabla_{\boldsymbol{\theta}} \widehat{\mathcal{L}}_t(\boldsymbol{\theta}_t, \boldsymbol{\lambda}_t) \right) \right\|_2^2 \\
&\leq \|\boldsymbol{\theta} - \boldsymbol{\theta}_t\|_2^2 + \eta^2 \left\| \nabla_{\boldsymbol{\theta}} \widehat{\mathcal{L}}_t(\boldsymbol{\theta}_t, \boldsymbol{\lambda}_t) \right\|_2^2 + 2\eta \left\langle \boldsymbol{\theta} - \boldsymbol{\theta}_t, \nabla_{\boldsymbol{\theta}} \widehat{\mathcal{L}}_t(\boldsymbol{\theta}_t, \boldsymbol{\lambda}_t) \right\rangle.
\end{aligned} \tag{c10}$$

Rearranging the terms in (c10) gives

$$\left\langle \boldsymbol{\theta}_t - \boldsymbol{\theta}, \nabla_{\boldsymbol{\theta}} \widehat{\mathcal{L}}_t(\boldsymbol{\theta}_t, \boldsymbol{\lambda}_t) \right\rangle \leq \frac{1}{2\eta} \left( \|\boldsymbol{\theta} - \boldsymbol{\theta}_t\|_2^2 - \|\boldsymbol{\theta} - \boldsymbol{\theta}_{t+1}\|_2^2 \right) + \frac{\eta}{2} \left\| \nabla_{\boldsymbol{\theta}} \widehat{\mathcal{L}}_t(\boldsymbol{\theta}_t, \boldsymbol{\lambda}_t) \right\|_2^2.$$

Then, we have

$$\begin{aligned}
\langle \boldsymbol{\theta}_t - \boldsymbol{\theta}, \nabla_{\boldsymbol{\theta}} \mathcal{L}(\boldsymbol{\theta}_t, \boldsymbol{\lambda}_t) \rangle \leq &\frac{1}{2\eta} \left( \|\boldsymbol{\theta} - \boldsymbol{\theta}_t\|_2^2 - \|\boldsymbol{\theta} - \boldsymbol{\theta}_{t+1}\|_2^2 \right) + \frac{\eta}{2} \left\| \nabla_{\boldsymbol{\theta}} \widehat{\mathcal{L}}_t(\boldsymbol{\theta}_t, \boldsymbol{\lambda}_t) \right\|_2^2 \\
&+ \left\langle \boldsymbol{\theta}_t - \boldsymbol{\theta}, \nabla_{\boldsymbol{\theta}} \mathcal{L}(\boldsymbol{\theta}_t, \boldsymbol{\lambda}_t) - \nabla_{\boldsymbol{\theta}} \widehat{\mathcal{L}}_t(\boldsymbol{\theta}_t, \boldsymbol{\lambda}_t) \right\rangle.
\end{aligned} \tag{c11}$$

Plugging (c9) and (c11) into (c7) yields

$$\begin{aligned}
\mathcal{L}(\boldsymbol{\theta}_t, \boldsymbol{\lambda}) - \mathcal{L}(\boldsymbol{\theta}, \boldsymbol{\lambda}_t) \leq &\frac{1}{2\eta} \left( \|\boldsymbol{\lambda} - \boldsymbol{\lambda}_t\|_2^2 - \|\boldsymbol{\lambda} - \boldsymbol{\lambda}_{t+1}\|_2^2 \right) + \frac{\eta}{2} \|\nabla_{\boldsymbol{\lambda}} \mathcal{L}(\boldsymbol{\theta}_t, \boldsymbol{\lambda}_t)\|_2^2 \\
&+ \frac{1}{2\eta} \left( \|\boldsymbol{\theta} - \boldsymbol{\theta}_t\|_2^2 - \|\boldsymbol{\theta} - \boldsymbol{\theta}_{t+1}\|_2^2 \right) + \frac{\eta}{2} \left\| \nabla_{\boldsymbol{\theta}} \widehat{\mathcal{L}}_t(\boldsymbol{\theta}_t, \boldsymbol{\lambda}_t) \right\|_2^2 \\
&+ \left\langle \boldsymbol{\theta}_t - \boldsymbol{\theta}, \nabla_{\boldsymbol{\theta}} \mathcal{L}(\boldsymbol{\theta}_t, \boldsymbol{\lambda}_t) - \nabla_{\boldsymbol{\theta}} \widehat{\mathcal{L}}_t(\boldsymbol{\theta}_t, \boldsymbol{\lambda}_t) \right\rangle.
\end{aligned} \tag{c12}$$

Summing (c12) over $t \in [T]$ yields

$$\begin{aligned}
\sum_{t=1}^{T} (\mathcal{L}(\boldsymbol{\theta}_t, \boldsymbol{\lambda}) - \mathcal{L}(\boldsymbol{\theta}, \boldsymbol{\lambda}_t)) \leq &\frac{1}{2\eta} \left( \|\boldsymbol{\lambda} - \boldsymbol{\lambda}_1\|_2^2 - \|\boldsymbol{\lambda} - \boldsymbol{\lambda}_{T+1}\|_2^2 \right) + \frac{\eta}{2} \sum_{t=1}^{T} \|\nabla_{\boldsymbol{\lambda}} \mathcal{L}(\boldsymbol{\theta}_t, \boldsymbol{\lambda}_t)\|_2^2 \\
&+ \frac{1}{2\eta} \left( \|\boldsymbol{\theta} - \boldsymbol{\theta}_1\|_2^2 - \|\boldsymbol{\theta} - \boldsymbol{\theta}_{T+1}\|_2^2 \right) + \frac{\eta}{2} \sum_{t=1}^{T} \left\| \nabla_{\boldsymbol{\theta}} \widehat{\mathcal{L}}_t(\boldsymbol{\theta}_t, \boldsymbol{\lambda}_t) \right\|_2^2 \\
&+ \sum_{t=1}^{T} \left\langle \boldsymbol{\theta}_t - \boldsymbol{\theta}, \nabla_{\boldsymbol{\theta}} \mathcal{L}(\boldsymbol{\theta}_t, \boldsymbol{\lambda}_t) - \nabla_{\boldsymbol{\theta}} \widehat{\mathcal{L}}_t(\boldsymbol{\theta}_t, \boldsymbol{\lambda}_t) \right\rangle.
\end{aligned} \tag{c13}$$

Since $\boldsymbol{\lambda}_1 = \mathbf{0}$, $\|\boldsymbol{\theta} - \boldsymbol{\theta}_1\|_2^2 \leq 4R^2$, Lemma C1 is proved by omitting the non-positive terms in (c13).

## C.2 Proof of Lemma C2

From the definition of $\nabla_{\boldsymbol{\lambda}}\mathcal{L}(\boldsymbol{\theta}, \boldsymbol{\lambda})$, for any $t \in [T]$, we have

$$\|\nabla_{\boldsymbol{\lambda}}\mathcal{L}(\boldsymbol{\theta}_t, \boldsymbol{\lambda}_t)\|_2^2 = \|\mathbf{g}(\boldsymbol{\theta}_t) - \delta\eta\boldsymbol{\lambda}_t\|_2^2 \le 2\|\mathbf{g}(\boldsymbol{\theta}_t)\|_2^2 + 2\delta^2\eta^2\|\boldsymbol{\lambda}_t\|_2^2 \overset{(a)}{\le} 2C^2 + 2\delta^2\eta^2\|\boldsymbol{\lambda}_t\|_2^2,$$

where $(a)$ is based on the boundedness of the constraint $\mathbf{g}(\boldsymbol{\theta})$ given in Assumption 3. Similarly, from the definition of $\nabla_{\boldsymbol{\theta}}\mathcal{L}(\boldsymbol{\theta}_t, \boldsymbol{\lambda}_t)$, for any $t \in [T]$, we have

$$\begin{aligned}
\|\nabla_{\boldsymbol{\theta}}\mathcal{L}(\boldsymbol{\theta}_t, \boldsymbol{\lambda}_t)\|_2^2 &= \left\|\nabla_{\boldsymbol{\theta}}\mathrm{PR}(\boldsymbol{\theta}_t) + \nabla_{\boldsymbol{\theta}}\mathbf{g}(\boldsymbol{\theta}_t)^\top\boldsymbol{\lambda}_t\right\|_2^2 \\
&\le 2\|\nabla_{\boldsymbol{\theta}}\mathrm{PR}(\boldsymbol{\theta}_t)\|_2^2 + 2\left\|\nabla_{\boldsymbol{\theta}}\mathbf{g}(\boldsymbol{\theta}_t)^\top\boldsymbol{\lambda}_t\right\|_2^2 \\
&\overset{(a)}{\le} 2L^2 + 2L_{\mathbf{g}}^2\|\boldsymbol{\lambda}_t\|_2^2,
\end{aligned}$$

where $(a)$ is based on the Lipschitz continuity of both the performative risk and the constraint. Then, we have

$$\begin{aligned}
\left\|\nabla_{\boldsymbol{\theta}}\widehat{\mathcal{L}}_t(\boldsymbol{\theta}_t, \boldsymbol{\lambda}_t)\right\|_2^2 &= \left\|\nabla_{\boldsymbol{\theta}}\mathcal{L}(\boldsymbol{\theta}_t, \boldsymbol{\lambda}_t) + \nabla_{\boldsymbol{\theta}}\widehat{\mathrm{PR}}_t(\boldsymbol{\theta}_t) - \nabla_{\boldsymbol{\theta}}\mathrm{PR}(\boldsymbol{\theta}_t)\right\|_2^2 \\
&\le 2\|\nabla_{\boldsymbol{\theta}}\mathcal{L}(\boldsymbol{\theta}_t, \boldsymbol{\lambda}_t)\|_2^2 + 2\left\|\nabla_{\boldsymbol{\theta}}\widehat{\mathrm{PR}}_t(\boldsymbol{\theta}_t) - \nabla_{\boldsymbol{\theta}}\mathrm{PR}(\boldsymbol{\theta}_t)\right\|_2^2 \\
&\le 4L^2 + 4L_{\mathbf{g}}^2\|\boldsymbol{\lambda}_t\|_2^2 + 2\left\|\nabla_{\boldsymbol{\theta}}\widehat{\mathrm{PR}}_t(\boldsymbol{\theta}_t) - \nabla_{\boldsymbol{\theta}}\mathrm{PR}(\boldsymbol{\theta}_t)\right\|_2^2.
\end{aligned}$$

By now, Lemma C2 is proved.

# D  Proof of Lemma 4

The proof of Lemma 4 will involve the accumulated parameter estimation error $\sum_{t=1}^{T}\left\|\widehat{\mathbf{A}}_t - \mathbf{A}\right\|_{\mathrm{F}}^2$, which is bounded by the follwing lemma.

**Lemma D1 (Parameter Estimation Error).** *Let* $\zeta_t = \frac{2}{\kappa_1(t-1)+2\kappa_3}$, $\forall t \in [T]$. *Under Assumption 5, the accumulated parameter estimation error is upper bounded by:*

$$\sum_{t=1}^{T}\mathbb{E}\left\|\widehat{\mathbf{A}}_t - \mathbf{A}\right\|_{\mathrm{F}}^2 \le \overline{\alpha}\ln(T),$$

*where* $\overline{\alpha} := \max\left\{\frac{2\kappa_3}{\kappa_1}\left\|\widehat{\mathbf{A}}_0 - \mathbf{A}\right\|_{\mathrm{F}}^2, \frac{8\kappa_2\,\mathrm{tr}(\boldsymbol{\Sigma})}{\kappa_1^2}\right\}$.

See § E for the proof. Next, we proceed to prove Lemma 4.

*Proof of Lemma 4.* To facilitate our analysis, we introduce a finite-sample approximation for the gradient $\nabla_{\boldsymbol{\theta}}\mathrm{PR}(\boldsymbol{\theta})$, defined as

$$\nabla_{\boldsymbol{\theta}}\widehat{\mathrm{PR}}(\boldsymbol{\theta}) := \frac{1}{n}\sum_{i=1}^{n}\left[\nabla_{\boldsymbol{\theta}}\ell\left(\boldsymbol{\theta}; Z_{0,i} + \mathbf{A}\boldsymbol{\theta}\right) + \mathbf{A}^\top\nabla_Z\ell\left(\boldsymbol{\theta}, Z_{0,i} + \mathbf{A}\boldsymbol{\theta}\right)\right].$$

Then, we have the following inequality:

$$\begin{aligned}
&\left\|\nabla_{\boldsymbol{\theta}}\widehat{\mathrm{PR}}_t(\boldsymbol{\theta}_t) - \nabla_{\boldsymbol{\theta}}\mathrm{PR}(\boldsymbol{\theta}_t)\right\|_2^2 \\
&\le 2\left\|\nabla_{\boldsymbol{\theta}}\widehat{\mathrm{PR}}_t(\boldsymbol{\theta}_t) - \nabla_{\boldsymbol{\theta}}\widehat{\mathrm{PR}}(\boldsymbol{\theta}_t)\right\|_2^2 + 2\left\|\nabla_{\boldsymbol{\theta}}\widehat{\mathrm{PR}}(\boldsymbol{\theta}_t) - \nabla_{\boldsymbol{\theta}}\mathrm{PR}(\boldsymbol{\theta}_t)\right\|_2^2.
\end{aligned} \tag{d1}$$

From Assumption 4, we have

$$\begin{aligned}
&\mathbb{E}\left\|\nabla_{\boldsymbol{\theta}}\widehat{\mathrm{PR}}(\boldsymbol{\theta}_t) - \nabla_{\boldsymbol{\theta}}\mathrm{PR}(\boldsymbol{\theta}_t)\right\|_2^2 \\
&\le \frac{1}{n^2}\sum_{i=1}^{n}\mathbb{E}_{Z_{0,i}\sim\mathcal{D}_0}\left\|\nabla_{\boldsymbol{\theta}}\ell\left(\boldsymbol{\theta}_t; Z_{0,i} + \mathbf{A}\boldsymbol{\theta}_t\right) + \mathbf{A}^\top\nabla_Z\ell\left(\boldsymbol{\theta}_t; Z_{0,i} + \mathbf{A}\boldsymbol{\theta}_t\right) - \nabla_{\boldsymbol{\theta}}\mathrm{PR}(\boldsymbol{\theta}_t)\right\|_2^2 \le \frac{\sigma^2}{n}.
\end{aligned}$$

The first term in (d1) is handled as follows. Plugging into the expression of $\nabla_{\boldsymbol{\theta}}\widehat{\mathrm{PR}}_t(\boldsymbol{\theta}_t)$ and $\nabla_{\boldsymbol{\theta}}\widehat{\mathrm{PR}}(\boldsymbol{\theta}_t)$, we have

$$
\left\|\nabla_{\boldsymbol{\theta}}\widehat{\mathrm{PR}}_t(\boldsymbol{\theta}_t) - \nabla_{\boldsymbol{\theta}}\widehat{\mathrm{PR}}(\boldsymbol{\theta}_t)\right\|_2^2
$$

$$
\leq \frac{2}{n^2}\sum_{i=1}^n \left\|\nabla_{\boldsymbol{\theta}}\ell\left(\boldsymbol{\theta}_t; Z_{0,i} + \widehat{\mathbf{A}}_t\boldsymbol{\theta}_t\right) - \nabla_{\boldsymbol{\theta}}\ell\left(\boldsymbol{\theta}_t; Z_{0,i} + \mathbf{A}\boldsymbol{\theta}_t\right)\right\|_2^2
$$

$$
+ \frac{2}{n^2}\sum_{i=1}^n \left\|\widehat{\mathbf{A}}_t^\top \nabla_Z\ell\left(\boldsymbol{\theta}_t; Z_{0,i} + \widehat{\mathbf{A}}_t\boldsymbol{\theta}_t\right) - \mathbf{A}^\top \nabla_Z\ell\left(\boldsymbol{\theta}_t; Z_{0,i} + \mathbf{A}\boldsymbol{\theta}_t\right)\right\|_2^2. \tag{d2}
$$

With the $\beta$-smoothness of the loss function given in Assumption 1, we have

$$
\frac{2}{n^2}\sum_{i=1}^n \left\|\nabla_{\boldsymbol{\theta}}\ell\left(\boldsymbol{\theta}_t; Z_{0,i} + \widehat{\mathbf{A}}_t\boldsymbol{\theta}_t\right) - \nabla_{\boldsymbol{\theta}}\ell\left(\boldsymbol{\theta}_t; Z_{0,i} + \mathbf{A}\boldsymbol{\theta}_t\right)\right\|_2^2 \leq \frac{2\beta^2}{n}\left\|\widehat{\mathbf{A}}_t - \mathbf{A}\right\|_{\mathrm{F}}^2 \|\boldsymbol{\theta}_t\|_2^2.
$$

Moreover, the last term in (d2) is bounded by

$$
\frac{2}{n^2}\sum_{i=1}^n \left\|\widehat{\mathbf{A}}_t^\top \nabla_Z\ell\left(\boldsymbol{\theta}_t; Z_{0,i} + \widehat{\mathbf{A}}_t\boldsymbol{\theta}_t\right) - \mathbf{A}^\top \nabla_Z\ell\left(\boldsymbol{\theta}_t; Z_{0,i} + \mathbf{A}\boldsymbol{\theta}_t\right)\right\|_2^2
$$

$$
\leq \frac{4}{n^2}\sum_{i=1}^n \left\|\widehat{\mathbf{A}}_t - \mathbf{A}\right\|_{\mathrm{F}}^2 \left\|\nabla_Z\ell\left(\boldsymbol{\theta}_t; Z_{0,i} + \widehat{\mathbf{A}}_t\boldsymbol{\theta}_t\right)\right\|_2^2
$$

$$
+ \frac{4\sigma_{\max}(\mathbf{A})}{n^2}\sum_{i=1}^n \left\|\nabla_Z\ell\left(\boldsymbol{\theta}_t; Z_{0,i} + \widehat{\mathbf{A}}_t\boldsymbol{\theta}_t\right) - \nabla_Z\ell\left(\boldsymbol{\theta}_t; Z_{0,i} + \mathbf{A}\boldsymbol{\theta}_t\right)\right\|_2^2
$$

$$
\overset{(a)}{\leq} \frac{4L_Z^2}{n}\left\|\widehat{\mathbf{A}}_t - \mathbf{A}\right\|_{\mathrm{F}}^2 + \frac{4\beta^2\sigma_{\max}(\mathbf{A})}{n}\left\|\widehat{\mathbf{A}}_t - \mathbf{A}\right\|_{\mathrm{F}}^2 \|\boldsymbol{\theta}_t\|_2^2,
$$

where $(a)$ is because the loss function is $\beta$-smooth and $L_Z$ Lipachitz continuous in $Z$. Plugging the above results into (d1) and taking expectation yields

$$
\mathbb{E}\left\|\nabla_{\boldsymbol{\theta}}\widehat{\mathrm{PR}}_t(\boldsymbol{\theta}_t) - \nabla_{\boldsymbol{\theta}}\mathrm{PR}(\boldsymbol{\theta}_t)\right\|_2^2 \leq \frac{2\sigma^2}{n} + \frac{4}{n}\left(2L_Z^2 + \beta^2 R^2\left(1 + 2\sigma_{\max}(\mathbf{A})\right)\right)\mathbb{E}\left\|\widehat{\mathbf{A}}_t - \mathbf{A}\right\|_{\mathrm{F}}^2,
$$

where we utilize the boundedness of the available set that $\|\boldsymbol{\theta}\|_2 \leq R, \forall \boldsymbol{\theta} \in \boldsymbol{\Theta}$. Summing the above inequality over $T$ iterations yields

$$
\sum_{t=1}^T \mathbb{E}\left\|\nabla_{\boldsymbol{\theta}}\widehat{\mathrm{PR}}_t(\boldsymbol{\theta}_t) - \nabla_{\boldsymbol{\theta}}\mathrm{PR}(\boldsymbol{\theta}_t)\right\|_2^2
$$

$$
\leq \frac{2T\sigma^2}{n} + \frac{4}{n}\left(2L_Z^2 + \beta^2 R^2\left(1 + 2\sigma_{\max}(\mathbf{A})\right)\right)\sum_{t=1}^T \mathbb{E}\left\|\widehat{\mathbf{A}}_t - \mathbf{A}\right\|_{\mathrm{F}}^2.
$$

Plugging into the result in Lemma D1 proves Lemma 4. $\qquad\square$

## E  Proof of Lemma D1

The proof of Lemma D1 utilizes the following two supporting lemmas.

**Lemma E1 (One-Step Improvement).** *Suppose that Assumption 5 holds. For any $t \in [T]$, choose stepsize $\zeta_t \in \left(0, \frac{2}{\kappa_3}\right)$. Then, the parameter estimates satisfy:*

$$
\mathbb{E}\left[\left.\left\|\widehat{\mathbf{A}}_t - \mathbf{A}\right\|_{\mathrm{F}}^2\right| \widehat{\mathbf{A}}_{t-1}\right] \leq \left(1 - \kappa_1\zeta_t\left(2 - \zeta_t\kappa_3\right)\right)\left\|\widehat{\mathbf{A}}_{t-1} - \mathbf{A}\right\|_{\mathrm{F}}^2 + 2\zeta_t^2\kappa_2 \operatorname{tr}(\boldsymbol{\Sigma}), \forall t \in [T].
$$

**Lemma E2 (Sequence Result).** *Consider a sequence $\{S_t\}_{t=1}^T$ satisfying*

$$
S_t \leq \left(1 - \frac{2}{t - 1 + t_0}\right)S_{t-1} + \frac{\alpha}{(t - 1 + t_0)^2}, \forall t \in [T],
$$

*where $t_0 \geq 0$ and $\alpha > 0$ are two constants. Then, we have*

$$
S_t \leq \frac{\max\{t_0 S_0, \alpha\}}{t + t_0}, \forall t \in [T].
$$

Proofs of Lemma E1 and Lemma E2 are respectively given in § E.1 and § E.2. With these two Lemmas, the proof of Lemma D1 is given below.

*Proof of Lemma D1.* For any $t \in [T]$, set $\zeta_t = \frac{2}{\kappa_1\left(t-1+\frac{2\kappa_3}{\kappa_1}\right)}$. Then, we have $2 - \zeta_t \kappa_3 = 2 - \frac{2\kappa_3}{\kappa_1(t-1)+2\kappa_3} \geq 1$. Plugging this inequality into Lemma E1, we have

$$\mathbb{E}\left[\left\|\widehat{\mathbf{A}}_t - \mathbf{A}\right\|_{\mathrm{F}}^2 \middle| \widehat{\mathbf{A}}_{t-1}\right] \leq \left(1 - \frac{2}{t-1+\frac{2\kappa_3}{\kappa_1}}\right)\left\|\widehat{\mathbf{A}}_{t-1} - \mathbf{A}\right\|_{\mathrm{F}}^2 + \frac{8\kappa_2 \operatorname{tr}(\boldsymbol{\Sigma})}{\kappa_1^2\left(t-1+\frac{2\kappa_3}{\kappa_1}\right)^2}.$$

Define $\overline{\alpha} := \max\left\{\frac{2\kappa_3}{\kappa_1}\left\|\widehat{\mathbf{A}}_0 - \mathbf{A}\right\|_{\mathrm{F}}^2, \frac{8\kappa_2 \operatorname{tr}(\boldsymbol{\Sigma})}{\kappa_1^2}\right\}$. By Lemma E2, we have

$$\mathbb{E}\left\|\widehat{\mathbf{A}}_t - \mathbf{A}\right\|_{\mathrm{F}}^2 \leq \frac{\overline{\alpha}}{t + \frac{2\kappa_3}{\kappa_1}}.$$

Summing the above inequality yields

$$\sum_{t=1}^{T} \mathbb{E}\left\|\widehat{\mathbf{A}}_t - \mathbf{A}\right\|_{\mathrm{F}}^2 \leq \sum_{t=1}^{T} \frac{\overline{\alpha}}{t + \frac{2\kappa_3}{\kappa_1}} \leq \overline{\alpha}\left(\ln\left(T + \frac{2\kappa_3}{\kappa_1}\right) - \ln\left(\frac{2\kappa_3}{\kappa_1}\right)\right) \leq \overline{\alpha}\ln(T),$$

which proves Lemma D1. $\qquad\qquad\square$

## E.1 Proof of Lemma E1

Denote by $\mathbf{b}_t := Z_t' - Z_t$. We have $\mathbb{E}[\mathbf{b}_t|\mathbf{u}_t] = \mathbf{A}\mathbf{u}_t$. Then,

$$\mathbb{E}\left[\|\mathbf{A}\mathbf{u}_t - \mathbf{b}_t\|_2^2 \middle| \mathbf{u}_t\right] = \operatorname{tr}\left(\mathbb{E}\left(\mathbf{A}\mathbf{u}_t - \mathbf{b}_t\right)(\mathbf{A}\mathbf{u}_t - \mathbf{b}_t)^\top \middle| \mathbf{u}_t\right) = 2\operatorname{tr}(\boldsymbol{\Sigma}). \qquad (\mathrm{e}1)$$

Recall that $\boldsymbol{\Sigma}$ is the variance of the base distribution $\mathcal{D}_0$. In Algorithm 1, the update rule of the parameter estimate is $\widehat{\mathbf{A}}_t = \widehat{\mathbf{A}}_{t-1} - \zeta_t\left(\widehat{\mathbf{A}}_{t-1}\mathbf{u}_t - \mathbf{b}_t\right)\mathbf{u}_t^\top$. Thus, we have

$$\left\|\widehat{\mathbf{A}}_t - \mathbf{A}\right\|_{\mathrm{F}}^2 = \left\|\widehat{\mathbf{A}}_{t-1} - \mathbf{A} - \zeta_t\left(\widehat{\mathbf{A}}_{t-1}^\top\mathbf{u}_t - \mathbf{b}_t\right)\mathbf{u}_t^\top\right\|_{\mathrm{F}}^2$$
$$= \left\|\widehat{\mathbf{A}}_{t-1} - \mathbf{A}\right\|_{\mathrm{F}}^2 - 2\zeta_t\left\langle\widehat{\mathbf{A}}_{t-1} - \mathbf{A}, \left(\widehat{\mathbf{A}}_{t-1}\mathbf{u}_t - \mathbf{b}_t\right)\mathbf{u}_t^\top\right\rangle + \zeta_t^2\left\|\left(\widehat{\mathbf{A}}_{t-1}\mathbf{u}_t - \mathbf{b}_t\right)\mathbf{u}_t^\top\right\|_{\mathrm{F}}^2.$$

Given $\widehat{\mathbf{A}}_{t-1}$ and $\mathbf{u}_t$, taking conditional expectation on the above equation gives

$$\mathbb{E}\left[\left\|\widehat{\mathbf{A}}_t - \mathbf{A}\right\|_{\mathrm{F}}^2 \middle| \widehat{\mathbf{A}}_{t-1}, \mathbf{u}_t\right]$$
$$= \left\|\widehat{\mathbf{A}}_{t-1} - \mathbf{A}\right\|_{\mathrm{F}}^2 - 2\zeta_t\left\langle\widehat{\mathbf{A}}_{t-1} - \mathbf{A}, \left(\widehat{\mathbf{A}}_{t-1}\mathbf{u}_t - \mathbb{E}[\mathbf{b}_t|\mathbf{u}_t]\right)\mathbf{u}_t^\top\right\rangle + \zeta_t^2\mathbb{E}\left[\left\|\left(\widehat{\mathbf{A}}_{t-1}\mathbf{u}_t - \mathbf{b}_t\right)\mathbf{u}_t^\top\right\|_{\mathrm{F}}^2 \middle| \widehat{\mathbf{A}}_{t-1}, \mathbf{u}_t\right]$$
$$= \left\|\widehat{\mathbf{A}}_{t-1} - \mathbf{A}\right\|_{\mathrm{F}}^2 - 2\zeta_t\left\|\left(\widehat{\mathbf{A}}_{t-1} - \mathbf{A}\right)\mathbf{u}_t\right\|_2^2 + \zeta_t^2\|\mathbf{u}_t\|_2^2\mathbb{E}\left[\left\|\widehat{\mathbf{A}}_{t-1}\mathbf{u}_t - \mathbf{b}_t\right\|_2^2 \middle| \widehat{\mathbf{A}}_{t-1}, \mathbf{u}_t\right]. \quad (\mathrm{e}2)$$

The term $\mathbb{E}\left[\left\|\widehat{\mathbf{A}}_{t-1}\mathbf{u}_t - \mathbf{b}_t\right\|_2^2 \middle| \widehat{\mathbf{A}}_{t-1}, \mathbf{u}_t\right]$ in (e2) satisfies

$$\mathbb{E}\left[\left\|\widehat{\mathbf{A}}_{t-1}\mathbf{u}_t - \mathbf{b}_t\right\|_2^2 \middle| \widehat{\mathbf{A}}_{t-1}, \mathbf{u}_t\right]$$
$$= \left\|\widehat{\mathbf{A}}_{t-1}\mathbf{u}_t - \mathbf{A}\mathbf{u}_t\right\|_2^2 + \mathbb{E}\left[\|\mathbf{A}\mathbf{u}_t - \mathbf{b}_t\|_2^2 \middle| \mathbf{u}_t\right] + 2\left\langle\widehat{\mathbf{A}}_{t-1}\mathbf{u}_t - \mathbf{A}\mathbf{u}_t, \mathbf{A}\mathbf{u}_t - \mathbb{E}[\mathbf{b}_t|\mathbf{u}_t]\right\rangle$$
$$\overset{(a)}{=} \left\|\left(\widehat{\mathbf{A}}_{t-1} - \mathbf{A}\right)\mathbf{u}_t\right\|_2^2 + 2\operatorname{tr}(\boldsymbol{\Sigma}). \qquad (\mathrm{e}3)$$

where $(a)$ is from (e1) and $\mathbb{E}[\mathbf{b}_t|\mathbf{u}_t] = \mathbf{A}\mathbf{u}_t$. Plugging (e3) into (e2) gives

$$\mathbb{E}\left[\left.\left\|\widehat{\mathbf{A}}_t - \mathbf{A}\right\|_{\mathrm{F}}^2\right|\widehat{\mathbf{A}}_{t-1}, \mathbf{u}_t\right] = \left\|\widehat{\mathbf{A}}_{t-1} - \mathbf{A}\right\|_{\mathrm{F}}^2 - 2\zeta_t \left\|\left(\widehat{\mathbf{A}}_{t-1} - \mathbf{A}\right)\mathbf{u}_t\right\|_2^2$$
$$+ \zeta_t^2 \|\mathbf{u}_t\|_2^2 \left\|\left(\widehat{\mathbf{A}}_{t-1} - \mathbf{A}\right)\mathbf{u}_t\right\|_2^2 + 2\zeta_t^2 \|\mathbf{u}_t\|_2^2 \operatorname{tr}(\mathbf{\Sigma}). \qquad \text{(e4)}$$

Taking conditional expectation over the random noise $\mathbf{u}_t$ gives

$$\mathbb{E}\left[\left.\|\mathbf{u}_t\|_2^2 \left\|\left(\widehat{\mathbf{A}}_{t-1} - \mathbf{A}\right)\mathbf{u}_t\right\|_2^2\right|\widehat{\mathbf{A}}_{t-1}\right] = \left\langle\left(\widehat{\mathbf{A}}_{t-1} - \mathbf{A}\right)\left(\widehat{\mathbf{A}}_{t-1} - \mathbf{A}\right)^\top, \mathbb{E}\left[\left.\|\mathbf{u}_t\|_2^2 \mathbf{u}_t\mathbf{u}_t^\top\right|\widehat{\mathbf{A}}_{t-1}\right]\right\rangle$$
$$\leq \kappa_3 \mathbb{E}\left[\left.\left\|\left(\widehat{\mathbf{A}}_{t-1} - \mathbf{A}\right)\mathbf{u}_t\right\|_2^2\right|\widehat{\mathbf{A}}_{t-1}\right]. \qquad \text{(e5)}$$

Plugging (e5) into (e4) yields

$$\mathbb{E}\left[\left.\left\|\widehat{\mathbf{A}}_t - \mathbf{A}\right\|_{\mathrm{F}}^2\right|\widehat{\mathbf{A}}_{t-1}\right]$$
$$\leq \left\|\widehat{\mathbf{A}}_{t-1} - \mathbf{A}\right\|_{\mathrm{F}}^2 - \left(2\zeta_t - \zeta_t^2 \kappa_3\right)\mathbb{E}\left[\left.\left\|\left(\widehat{\mathbf{A}}_{t-1} - \mathbf{A}\right)\mathbf{u}_t\right\|_2^2\right|\widehat{\mathbf{A}}_{t-1}\right] + 2\zeta_t^2 \kappa_2 \operatorname{tr}(\mathbf{\Sigma}).$$

Further, we have

$$\mathbb{E}\left[\left.\left\|\left(\widehat{\mathbf{A}}_{t-1} - \mathbf{A}\right)\mathbf{u}_t\right\|_2^2\right|\widehat{\mathbf{A}}_{t-1}\right] = \operatorname{tr}\left(\left(\widehat{\mathbf{A}}_{t-1} - \mathbf{A}\right)^\top\left(\widehat{\mathbf{A}}_{t-1} - \mathbf{A}\right)\mathbb{E}\left[\left.\mathbf{u}_t\mathbf{u}_t^\top\right|\widehat{\mathbf{A}}_{t-1}\right]\right) \geq \kappa_1 \left\|\widehat{\mathbf{A}}_{t-1} - \mathbf{A}\right\|_{\mathrm{F}}^2.$$

Then, choosing $\zeta_t \in \left(0, \frac{2}{\kappa_3}\right)$, we obtain

$$\mathbb{E}\left[\left.\left\|\widehat{\mathbf{A}}_t - \mathbf{A}\right\|_{\mathrm{F}}^2\right|\widehat{\mathbf{A}}_{t-1}\right] \leq \left(1 - \kappa_1\zeta_t\left(2 - \zeta_t\kappa_3\right)\right)\left\|\widehat{\mathbf{A}}_{t-1} - \mathbf{A}\right\|_{\mathrm{F}}^2 + 2\zeta_t^2 \kappa_2 \operatorname{tr}(\mathbf{\Sigma}),$$

which proves Lemma E1.

### E.2 Proof of Lemma E2

We prove Lemma E2 by induction. First, for $t = 0$, $S_0 \leq \frac{\max\{t_0 S_0, \alpha\}}{t_0}$ automatically holds. Define $\overline{\alpha} := \max\{t_0 S_0, \alpha\}$. Suppose that $S_{t-1} \leq \frac{\overline{\alpha}}{t-1+t_0}$ holds. Then, we have

$$S_t \leq \left(1 - \frac{2}{t-1+t_0}\right)S_t + \frac{\alpha}{(t-1+t_0)^2}$$
$$\leq \left(1 - \frac{2}{t-1+t_0}\right)\frac{\overline{\alpha}}{t-1+t_0} + \frac{\alpha}{(t-1+t_0)^2}$$
$$\leq \frac{\overline{\alpha}}{t-1+t_0} - \frac{2\overline{\alpha}}{(t-1+t_0)^2} + \frac{\overline{\alpha}}{(t-1+t_0)^2}$$
$$\leq \frac{\overline{\alpha}}{t-1+t_0} - \frac{\overline{\alpha}}{(t-1+t_0)^2} \leq \frac{\overline{\alpha}}{t+t_0},$$

where the last inequality is based on the fact that $(t+t_0)(t-2+t_0) = (t-1-t_0)^2 - 1 \leq (t-1-t_0)^2$. Thus, $\frac{1}{t-1+t_0} - \frac{1}{(t-1+t_0)^2} = \frac{t-2+t_0}{(t-1-t_0)^2} \leq \frac{1}{t+t_0}$. By now, we have proved Lemma E2.

## F  Experiment Details

In this section, we elaborate on the simulation details of the numerical experiments in Section 5.

### F.1  Multi-Task Linear Regression

The multi-task linear regression is conducted over a randomly generated Erdos-Renyi graph with $n = 10$ nodes. The probability of an edge between any pair of nodes in the Erdos-Renyi graph

is 0.5. The decision dimension of each task is set to be 3. The decision of task $i$ is initialized as $\boldsymbol{\theta}_i = \mathbf{0}, \forall i \in \mathcal{V}$. The injected noises $\{\mathbf{u}_t\}_{t=1}^T$ are independently drawn from $\mathcal{N}(\mathbf{0}, \mathbf{I})$. The number of iterations is $T = 10^6$. The number of initial samples is $n = 10^3$. The stepsize of the alternating gradient update is $\eta = 5 \times 10^{-3}$. The control parameter is $\delta = 1$. The stepsize of the online parameter estimation at the $t$th iteration is $\zeta_t = \frac{1}{t+10}, \forall t \in [T]$.

**Data Generation Process:** For any $i \in \mathcal{V}$, given a parameter vector $\boldsymbol{\theta}_i \in \mathbb{R}^d$, the feature-label pair $(\mathbf{x}_i, y_i)$ is generated as follows:

1. $\mathbf{x}_i \sim \mathcal{N}(\mathbf{0}, \boldsymbol{\Sigma}_{\mathbf{x}_i})$, where $\boldsymbol{\Sigma}_{\mathbf{x}_i}$ is a random symmetric positive-definite matrix with nuclear norm $d$.

2. $y_i = \boldsymbol{\beta}_i^\top \mathbf{x}_i + \boldsymbol{\mu}_i^\top \boldsymbol{\theta}_i + w_i$, where $\boldsymbol{\beta}_i \sim \mathcal{N}(\mathbf{0}, \mathbf{I})$ and $w_i \sim \mathcal{N}(0, \sigma_i^2)$ with $\sigma_i^2 = 1$.

This distribution map is a location family with sensitivity parameter $\varepsilon = \sum_{i \in \mathcal{V}} \|\boldsymbol{\mu}_i\|_2$. To generate all the vectors $\{\boldsymbol{\mu}_i\}_{i \in \mathcal{V}}$, we first independently draw $|\mathcal{V}|$ samples from $\mathcal{N}(\mathbf{0}, \mathbf{I})$ and then projected their concatenation onto the sphere of radius $\varepsilon$.

In constraint-free case, given the squared-loss $\ell_i(\boldsymbol{\theta}_i; (\mathbf{x}_i, y_i)) = \frac{1}{2}(y_i - \boldsymbol{\theta}_i^\top \mathbf{x}_i)^2$ and the linearity of the performative effect, the performative optimum of each task $i$, denoted by $\boldsymbol{\theta}_{i,\mathrm{PO}}$, can be computed in closed-form as

$$\boldsymbol{\theta}_{i,\mathrm{PO}} = \mathcal{C}_{x_i x_i}^{-1} \mathcal{C}_{x_i y_i}, \forall i \in \mathcal{V},$$

where $\mathcal{C}_{x_i x_i} := \boldsymbol{\Sigma}_{\mathbf{x}_i} + \boldsymbol{\mu}_i \boldsymbol{\mu}_i^\top$ and $\mathcal{C}_{x_i y_i} := \boldsymbol{\Sigma}_{\mathbf{x}_i} \boldsymbol{\beta}_i, \forall i \in \mathcal{V}$. Correspondingly, the minimum performative risk is given by

$$\mathrm{PR}(\boldsymbol{\theta}_{\mathrm{PO}}) = \sum_{i \in \mathcal{V}} \mathcal{C}_{y_i y_i} - \mathcal{C}_{y_i x_i} \mathcal{C}_{x_i x_i}^{-1} \mathcal{C}_{x_i y_i},$$

where $\boldsymbol{\theta}_{\mathrm{PO}}$ is the concatenation of $\boldsymbol{\theta}_{i,\mathrm{PO}}$ for all $i \in \mathcal{V}$, $\mathcal{C}_{y_i y_i} := \boldsymbol{\beta}_i^\top \boldsymbol{\Sigma}_{\mathbf{x}_i} \boldsymbol{\beta}_i + \sigma_i^2$, $\mathcal{C}_{y_i x_i} = \mathcal{C}_{x_i y_i}^\top$.

The constraint associated with each neighboring node pair $(i, j) \in \mathcal{E}$ is set to be

$$\|\boldsymbol{\theta}_i - \boldsymbol{\theta}_j\|_2^2 \leq \|\boldsymbol{\theta}_{i,\mathrm{PO}} - \boldsymbol{\theta}_{j,\mathrm{PO}}\|_2^2 + (b_{ij}')^2,$$

where $\{b_{ij}'\}_{(i,j) \in \mathcal{E}}$ are uniformly drawn from the region $[0, 0.02]$ in a symmetry manner, i.e., $b_{ij}' = b_{ji}'$, $\forall (i, j) \in \mathcal{E}$.

Let $\boldsymbol{\theta}$ be the concatenation of $\boldsymbol{\theta}_i$ for all $i \in \mathcal{V}$. The approximate performative gradient of APDA is computed by

$$\nabla_{\boldsymbol{\theta}} \widehat{\mathrm{PR}}_t(\boldsymbol{\theta}_t) = \frac{1}{n} \sum_{i \in \mathcal{V}} \sum_{j=1}^n \left[ \left( y_{i,j} - \boldsymbol{\theta}_{i,t}^\top \mathbf{x}_{i,j} \right) \left( \widehat{\boldsymbol{\mu}}_{i,t} - \mathbf{x}_{i,j} \right) \right], \forall t \in [T].$$

The approximate performative gradient of PD-PS is computed by

$$\nabla_{\boldsymbol{\theta}} \widehat{\mathrm{PR}}_t(\boldsymbol{\theta}_t) = -\frac{1}{n} \sum_{i \in \mathcal{V}} \sum_{j=1}^n \left[ \left( y_{i,j} - \boldsymbol{\theta}_{i,t}^\top \mathbf{x}_{i,j} \right) \mathbf{x}_{i,j} \right], \forall t \in [T].$$

The approximate performative gradient of the "baseline" is computed by

$$\nabla_{\boldsymbol{\theta}} \mathrm{PR}_t(\boldsymbol{\theta}_t) = \frac{1}{n} \sum_{i \in \mathcal{V}} \sum_{j=1}^n \left[ \left( y_{i,j} - \boldsymbol{\theta}_{i,t}^\top \mathbf{x}_{i,j} \right) \left( \boldsymbol{\mu}_i - \mathbf{x}_{i,j} \right) \right], \forall t \in [T].$$

The performative risk is computed by

$$\mathrm{PR}(\boldsymbol{\theta}_t) = \mathcal{C}_{y_i y_i} - \mathcal{C}_{y_i x_i} \boldsymbol{\theta}_t - \boldsymbol{\theta}_t^\top \mathcal{C}_{x_i y_i} + \boldsymbol{\theta}_t^\top \mathcal{C}_{x_i x_i}^{-1} \boldsymbol{\theta}_t, \forall t \in [T].$$

All results are averaged over 100 realizations.

### F.2 Multi-Asset Portfolio

In the implementation of the multi-asset portfolio, we add a regularizer $\xi\|\boldsymbol{\theta}\|_2^2$ to the original loss function to make it strongly convex. This gives the optimization problem:

$$\min_{\boldsymbol{\theta}} \quad - \mathop{\mathbb{E}}_{\mathbf{z}\sim\mathcal{D}(\boldsymbol{\theta})} \mathbf{z}^\top\boldsymbol{\theta} + \xi\|\boldsymbol{\theta}\|_2^2$$

$$\text{s.t.} \quad \sum_{i=1}^{l}\theta_i \leq 1,$$

$$\mathbf{0} \preceq \boldsymbol{\theta} \preceq \epsilon\cdot\mathbf{1},$$

$$\mathbf{s}^\top\boldsymbol{\theta} \leq S,$$

$$\boldsymbol{\theta}^{\mathrm{T}}\boldsymbol{\Psi}\boldsymbol{\theta} \leq \rho.$$

In the simulation, we set the number of assets $l = 10$. The initial investment decision $\boldsymbol{\theta}_1$ is randomly chosen within the feasible set. The injected noises $\{\mathbf{u}_t\}_{t=1}^{T}$ are independently drawn from $\mathcal{N}(\mathbf{0}, \mathbf{I})$. The parameter $\xi$ in the regularizer is set to be $\varepsilon$. The maximum amount of investment to one asset is $\epsilon = 0.3$. The entries of the bid-ask spread vector $\mathbf{s}$ are independently and uniformly drawn from the region $[2, 4]$. The maximum allowable bid-ask spread is $S = 2$. The risk tolerance threshold is $\rho = 0.01$. The number of iterations is $T = 10^6$. The number of initial samples is $n = 10^3$. The stepsize of the alternating gradient update is $\eta = 5 \times 10^{-3}$. The control parameter is $\delta = 1$. The stepsize of the online parameter estimation at the $t$th iteration is $\zeta_t = \frac{1}{t+10}, \forall t \in [T]$.

**Data Generation Process:** The rate of reture follows $\mathbf{z} = \bar{\mathbf{z}} + \mathbf{A}\boldsymbol{\theta} + \mathbf{u}_{\mathbf{z}}$, where $\bar{\mathbf{z}}$ is a constant vector, $\mathbf{u}_{\mathbf{z}} \sim \mathcal{N}(\mathbf{0}, \boldsymbol{\Sigma}_{\mathbf{z}})$, and $\boldsymbol{\Sigma}_{\mathbf{z}}$ is a random symmetric positive-definite matrix with nuclear norm $1/l$. To generate $\bar{\mathbf{z}}$, we first uniformly draw a sample within the region $[10\varepsilon, 1 + 10\varepsilon]$ and then project it onto the sphere of radius 2.

This distribution map is a location family with sensitivity parameter $\varepsilon = \sigma_{\max}(\mathbf{A})$. Optimization of the multi-asset portfolio problem requires the covariance matrix of $\mathbf{z}$, which is unknown. Note that the randomness of $\mathbf{z}$ lies in the term $\mathbf{u}_{\mathbf{z}}$. Then, we have $\boldsymbol{\Psi} = \boldsymbol{\Sigma}_{\mathbf{z}}$. The covariance matrix $\boldsymbol{\Sigma}_{\mathbf{z}}$ can be approximated based on the initial samples drawn from $\mathcal{D}(\mathbf{0})$. The optimal investment is computed by CVX tools [Grant and Boyd, 2014].

The approximate performative gradient of APDA is given by

$$\nabla_{\boldsymbol{\theta}}\widehat{\mathrm{PR}}_t(\boldsymbol{\theta}_t) = -\frac{1}{n}\sum_{i=1}^{l}\sum_{j=1}^{n}\mathbf{z}_j + \left(2\xi\cdot\mathbf{I} - \widehat{\mathbf{A}}_t\right)\boldsymbol{\theta}, \forall t \in [T].$$

The approximate performative gradient of PD-PS is given by

$$\nabla_{\boldsymbol{\theta}}\widehat{\mathrm{PR}}_t(\boldsymbol{\theta}_t) = -\frac{1}{n}\sum_{i=1}^{l}\sum_{j=1}^{n}\mathbf{z}_j + 2\xi\boldsymbol{\theta}, \forall t \in [T].$$

The approximate performative of the "baseline" is given by

$$\nabla_{\boldsymbol{\theta}}\widehat{\mathrm{PR}}_t(\boldsymbol{\theta}_t) = -\frac{1}{n}\sum_{i=1}^{l}\sum_{j=1}^{n}\mathbf{z}_j + (2\xi\cdot\mathbf{I} - \mathbf{A})\boldsymbol{\theta}, \forall t \in [T].$$

The performative risk is given by

$$\mathrm{PR}(\boldsymbol{\theta}_t) = \bar{\mathbf{z}}^\top\boldsymbol{\theta}_t + \boldsymbol{\theta}_t^\top\mathbf{A}\boldsymbol{\theta}_t + \xi\|\boldsymbol{\theta}_t\|_2^2.$$

All results are averaged over 100 realizations.