# OpenReview forum: "Zero-Regret Performative Prediction Under Inequality Constraints"
_NeurIPS.cc/2023/Conference — NeurIPS 2023 poster_

### Official Review · Reviewer_9Way · 2023-06-14

**Soundness:** 4 excellent
**Presentation:** 3 good
**Contribution:** 2 fair
**Rating:** 4
**Confidence:** 4

**Summary:**

The paper provides a new algorithm for performative prediction under general inequality constraints. The proposed algorithm has sublinear regret with respect to the performative optimum if the distribution map follows the location family model. The theory is supported by simulations.

**Strengths:**

The motivation behind introducing constraints in performative prediction is strong and justified. The explicit regret guarantees are compelling. I thought Example 1 was well-motivated, clear, and a great example to keep in mind throughout. Section 3 is well-written and gives a clear step-by-step outline of the main method. The connection to existing work is thorough.

**Weaknesses:**

The novelty has limitations. The main method combines existing ideas in the literature quite directly. The method for constructing a zeroth-order estimate of the gradient follows the same recipe as the two-stage method of Miller et al., and in a way this is the heart of the proposed method. This I see as the most relevant weakness. Below I'm including additional points.

It is not clear what the meaning of the time horizon T is in your analyses. Normally this should be the number of collected samples. But in your paper this is not the case, and I think that for this reason your T has a somewhat arbitrary meaning. I would rewrite the result statements to make T the total number of collected samples (I don't think this should change your rates). This is important especially because when you tune n you set it to grow with T as sqrt(T).

Assumption 1 is just there to ensure convexity of the objective, right? If so, maybe it's cleaner to state that assumption directly, and give the conditions in Assumption 1 as one way to satisfy the convexity condition. Otherwise, the way things are stated right now, your Example 1 wouldn't be handled by the assumptions?

In Lemma 3 (and other results that rely on this lemma) I would probably write something like "there exists a delta(\eta, L_g) such that..." Right now the interval for delta is too cumbersome.

You mention that derivative-free methods such as Flaxman et al. could be used for your problem. I think it would be a good idea to spell out exactly what they give you, because although you would lose dimension-dependent factors the results would be applicable much more broadly (all convex problems, not just location families). Location families are interesting but they are too specific.



**Questions:**

In line 45, when you say "exponential structure" you mean exponential families?

Line 148, denoted -> denote.

I don't understand why you have both eta and delta. In problem (2) they appear together. I don't understand why they are tuned separately in the end? Why are they not treated as a single parameter?

The notation in Algorithm 1 is strange. What do you mean by Z_t \sim \D(Z_0 + A-hat_t theta_t)? Also, shouldn't it be the true A and not A_t-hat?

Please define PD-PS used in line 306.

Line 320, sensitive -> sensitivity.

Line 323 says accuracy decreases. You mean error decreases?

---

> ### Author Rebuttal · Authors · 2023-08-04
>
> Thanks a lot for your review and for recognizing our contribution to the research field of performative prediction. Our point-to-point responses to concerns on weaknesses and questions are given below:
>
> **Reply to Weaknesses:**
>
> **1.** Thank you for the comment. The main contributions of this paper are listed below:
>
> - We are the first to study the performative prediction problem under inequality constraints, which is ubiquitous in the real world but currently understudied.
> - We developed a robust primal-dual framework that requires only approximate gradients up to an accuracy of $\mathcal{O}(\sqrt{T})$, yet delivers the same order of performance as the stochastic primal-dual algorithm without performativity.
> - Based on this framework, we propose an adaptive primal-dual algorithm for location families, which achieves $\mathcal{O}(\sqrt{T})$ regret and constraint violations, using only $\sqrt{T} + 2T$ samples.
>
> We understand that the reviewer is suggesting that our adaptive parameter estimation for the location family bears some resemblance to the two-stage algorithm in [Miller et al., 2021]. However, they are different approaches. More importantly, we would like to emphasize that parameter estimation does not constitute the main contribution of this paper. The location family serves as a mere application example of our robust primal-dual framework, which can be applied to other distribution forms with effective gradient approximation methods. Our contributions have been recognized by other reviewers, as cited below.
>
> - PDku: “This paper studies and analyze the optimality of the performative prediction problem under inequality constraints, which is an important yet missing piece in the literature. The proposed robust primal-dual method is intuitive. The theoretical analysis is throughout and it also provides empirical study to justify the findings.”
> - NMva: “The proposed primal-dual algorithm is very nice in the sense that (i) the gradient estimators don't need to be unbiased; and (ii) the authors prove a regret bound in terms of the accumulated gradient approximation error. This can be a valuable contribution to the research community.”
>
> **2.**  Thank you for the comment.  In our algorithm, the time horizon $T$ refers to the number of iterations or decisions taken. The term "time horizon" follows the field of online optimization. We employ the number of iterations in our convergence analysis because it better reflects the convergence speed of an optimization algorithm. Generally, the convergence speed is of more concern than the number of samples when evaluating the rate of convergence of an algorithm. We agree with the reviewer that changing $T$ to the number of samples will not affect the rate of our algorithm since the required samples are in the same order as the number of iterations.
>
> **3.**  $1)$ Thank you for the comment. Assumption 1 pertains to the properties of the loss function, which is commonly made in the literature of performative prediction. Under Assumption 1, we demonstrate the convexity of the performative risk for distribution maps within the location family. Directly assuming the convexity of the performative risk is uncommon in the literature, and hence, we do not do so.
>
> $2)$  Example 1 serves to demonstrate a practical application of performative prediction with constraints. The loss function of Example 1 does not need to meet the assumptions of our algorithm. To apply our algorithm to Example 1, we can add a regularizer $||\theta||_2^2$ on its loss function to make it strongly convex. This approach is widely used in the literature, such as in [Narang et al., 2022] and [Mandal et al., 2022].
>
> **4.**  Thank you for the helpful comment! We have changed the related descriptions throughout our paper according to this comment.
>
> **5.**  Thank you for the comment. Derivative-free methods approximate the gradient $\nabla_\theta PR (\theta)$ by querying the objective at random points around the current iterate. For example, by the derivative-free method in [Flaxman et al. (2005)], the gradient $\nabla_\theta PR (\theta)$ can be approximated as $\nabla_\theta PR (\theta) \approx \frac{d}{\xi} \mathbb{E}PR(\theta + \xi u)u$. The analysis results in [Flaxman et al. (2005)] demonstrate that the convergence bound of this method is in $\mathcal{O}(d^2)$ w.r.t. the data dimension $d$.
>
>
> **Reply to Questions:**
>
> **1.**  Thank you for the question. As you correctly pointed out, "exponential structure" in line 45 refers to exponential families. To avoid misunderstanding, we have clarified this in the revised manuscript.
>
> **2.**  Thank you for the helpful comment!  We have rectified this typo.
>
> **3.**  Thank you for the question. $\eta$ is the stepsize of the algorithm, and $\delta$ is a control parameter that determines the strength of the regularizer $-\frac{\delta\eta}{2}||\lambda||^2$. Both parameters appear in the regularizer term for convergence analysis purposes. We want to carefully design the values of these two parameters to eliminate some $\lambda$-related terms in the convergence bound. This careful selection is necessary to improve the efficiency of our algorithm.
>
> **4.**  Thank you for the question. We have revised the notation to $Z_t\sim D(\theta_t)$, which means that the sample $Z_t$ is drawn from the distribution $D(\theta_t)$.
>
> **5.**  Thank you for the helpful comment! We have included a more detailed description of the PD-PS algorithm in our manuscript. PD-PS stands for the primal-dual algorithm used to find the performative stable points. The algorithm is similar to APDA but uses only the first term in Eq. (8) as the approximate gradient.
>
> **6.**  Thank you for the helpful comment! We have rectified this typo.
>
> **7.**  Thank you for the helpful comment! We have revised the description.
>
>
> **We would be happy to discuss with the reviewer should there be any other concerns/questions. Thank you very much!**

---

> > ### Comment · Reviewer_9Way · 2023-08-12
> > **Post response**
> >
> > Thank you for the responses! They were clarifying.

---

> > > ### Author Response · Authors · 2023-08-13
> > >
> > > Dear Reviewer 9Way:
> > >
> > > Thank you for dedicating your time to reviewing our paper and providing valuable feedback. We greatly appreciate your efforts in assisting us with improving our work.
> > >
> > > We were pleased to learn that you found our responses clarifying. However, we noticed that our paper was still rated as a borderline reject in your review. We kindly request you to inform us of any additional concerns that lead to this rating. We would be grateful if you could reconsider your evaluation by taking another look at our paper. We have made significant improvements based on the feedback received from all the reviewers. Our contributions have been acknowledged and recognized by other reviewers as meeting the standards of NeurIPS, as evidenced by the following citations:
> > >
> > > •	Reviewer "Puku": "This paper addresses the performative prediction problem under inequality constraints, which is an important yet unexplored area in the literature. To the best of my knowledge, this is the first paper to propose a solution to this problem. The paper is well-written, easy to follow, and the proposed robust primal-dual method is intuitive. The theoretical analysis is comprehensive and supported by empirical studies."
> > >
> > > •	Reviewer "NMVA": "The proposed primal-dual algorithm is impressive because (i) the gradient estimators do not need to be unbiased, and (ii) the authors establish a regret bound based on the accumulated gradient approximation error. This contribution can be highly valuable to the research community."
> > >
> > > Thank you for your consideration, and we eagerly await your response.
> > >
> > > Best regards,
> > >
> > > Authors of the paper.

---

> > > > ### Comment · Reviewer_9Way · 2023-08-16
> > > > **Followup**
> > > >
> > > > Thank you for following up. You have answered my questions and clarified some points, and I appreciate that. As I indicated in my initial review, the soundness and presentation of the paper are very good, so clarity was never the main issue. The main reason for my score not being higher is my perception of the paper's conceptual and technical novelty.

---

> > > > > ### Author Response · Authors · 2023-08-17
> > > > >
> > > > > Dear reviewer 9Way:
> > > > >
> > > > > Thank you for your reply. We appreciate your recognition on the soundness and presentation of this paper. While we respect your decision, we would like to provide additional explanations for your concern about the novelty of this paper.
> > > > >
> > > > > In light of your comments, the main concern about the novelty of our paper is the similarity of the parameter estimation method for the distribution maps with the method in [Miller et al., 2021], as evidenced by the citation below:
> > > > > >The main method combines existing ideas in the literature quite directly. The method for constructing a zeroth-order estimate of the gradient follows the same recipe as the two-stage method of Miller et al., and in a way this is the heart of the proposed method. This I see as the most relevant weakness.
> > > > >
> > > > > However, it is important to note that our paper differs significantly from the work [Miller et al., 2021]. Firstly, we study different problems: [Miller et al., 2021] address unconstrained problems while we focus on performative prediction under inequality constraints. Second, we propose different algorithms. In [Miller et al., 2021], the authors proposed to minimize a finite-sample approximation of the performative risk, but they did not give any specific algorithm to solve this minimization problem.
> > > > >
> > > > >  In our paper, to solve the constrained performative prediction problem, we first developed a robust primal-dual framework that requires only approximate gradients up to an accuracy of $\mathcal{O}(\sqrt{T})$, yet delivers the same order of performance as the stochastic primal-dual algorithm without performativity. Notably, the robust primal-dual framework does not restrict the approximate gradients to be unbiased and hence offers more flexibility to the design of gradient approximation. Based on this framework, we propose an adaptive primal-dual algorithm for location families, which consists of an online stochastic approximation and an offline parameter estimation for the performative gradient approximation. Our analysis demonstrates that the proposed algorithm achieves $\mathcal{O}(\sqrt{T})$ regret and constraint violations, using only$\sqrt{T}+2T$ sample.
> > > > >
> > > > > The sole similarity between these two algorithms lies in the parameter estimation method, both of which are based on least squares. However, there are some differences between the two approaches. Firstly, our parameter estimation method is online, and its accuracy improves with the iteration of the primal-dual algorithm. In contrast, in [Miller et al., 2021], the parameter estimation and the performative risk minimization are conducted in two separate stages. Secondly, our online least-squares are based on the injected random noises, while Miller’s is based on the observed samples.
> > > > >
> > > > > Actually, in the study on performative optimality, estimating the underlying distribution maps is almost unavoidable for anticipating the performative gradient. To our best knowledge, the only paper on performative optimality that does not need parameter estimation is [Jagadeesan et al., 2022], of which the key idea is to exhaustively explore the feasible region with an efficient discarding mechanism. However, the sample complexity of the algorithm in [Jagadeesan et al., 2022] is $\mathcal{O}(T\log(T))$. In contrast, our algorithm only requires a total of $\sqrt{T}+2T$ samples.
> > > > >
> > > > > More importantly, as we have mentioned in the previous response, parameter estimation does not constitute the main contribution of this paper. The location family serves as a mere application example of our robust primal-dual framework, which can be applied to other distribution forms with effective gradient approximation methods. For instance, the exponential family considered in [Izzo et al., 2021] with their gradient approximation method can be directly applied to our robust primal-dual framework. Our contributions are in the first study on the constrained performative prediction problem and the development of the robust primal-dual algorithm to find the optimal point for the problem.
> > > > >
> > > > > We hope that the above explanations can address your concern. Thank you once again for your review.
> > > > >
> > > > > Best regards,
> > > > >
> > > > > Authors of the paper.

---

### Official Review · Reviewer_yrZY · 2023-07-04

**Soundness:** 2 fair
**Presentation:** 2 fair
**Contribution:** 2 fair
**Rating:** 5
**Confidence:** 2

**Summary:**

The paper attempts to study the problem of learning where the problem instances arise from a distribution. What is special is that this distribution is dependent on the decision-maker. The specific problem is when the distribution is given by a linear shift over a fixed distribution where the degree of shift is given by the decision parameters itself, and the decision parameters itself need to satisfy an inequality constraint. The authors propose a PD algorithm that  (under certain assumptions on the loss function) queries $T+2\cdot\sqrt{T}$ samples and violates $\sqrt{T}$ constraints to yield a $O(\sqrt{T})$ regret across $T$ rounds.


**Strengths:**

The paper makes a good attempt at trying to look at the optimization problem with the inclusion of inequality constraints and it able to consider two simultaneous performance measures, namely, regret and (extent of) constraint violations.

**Weaknesses:**

The paper promises a study in the direction of general optimization under (decision dependent) uncertainty but only tackles the problem in the limited case where the distribution is linear in the decision space. There is no discussion (or any insight) on how these techniques may generalize for more complicated distributions. The algorithm design itself does not appear to be very novel and uses fairly standard statistical estimation techniques.

**Questions:**

How is Eq (4) (gradient of $PR(\theta)$ wrt $\theta$ ) on Pg 4 derived from Eq(1) (Definition of $PR(\theta)$)?

Is there any hope (or negative results) in the case where the distribution does not follow a linear behavior wrt to decision parameter ? I would expect the sample complexity to rise wrt to the complexity of the dependence b/w $\theta$ and the distribution function $D(\theta)$.

This body of work sounds similar to areas where the prediction problem is different from the optimization problem i.e. areas where the performance measure depends on both the unknown problem parameter as well as algorithm design (decision parameter), and the algorithm itself makes decision based on this caveat.
Examples:
1. Customizing ML predictions for online algorithms (Anand et al)
2. Learning Predictions for Algorithms with Predictions (Khodak et al)
Can you draw any parallels or is there no connection?

**Limitations:**

Authors have addressed limitation adequately.

---

> ### Author Rebuttal · Authors · 2023-08-03
>
> Thank you for your review and for recognizing our contribution to the research field of performative prediction. Our point-to-point responses to concerns on weaknesses and questions are given below:
>
> **Reply to Weaknesses:**
>
> 1.	> The paper promises a study in the direction of general optimization under (decision dependent) uncertainty but only tackles the problem in the limited case where the distribution is linear in the decision space. There is no discussion (or any insight) on how these techniques may generalize for more complicated distributions.
>
> **Reply:** Thank you for the comment . As replied to Weakness 2 of the reviewer "PDku", our robust primal-dual framework is not limited to any distribution forms. The location family merely serves as an application example, and our framework can be applied to other distribution forms with effective gradient approximation methods. For instance, the exponential family considered in [Izzo et al., 2021] with their gradient approximation method can be directly applied to our robust primal-dual framework.
>
> We thank the reviewer for pointing this out. We have clarified the generalizability of our robust primal-dual framework to other distributions in the revised manuscript.
>
> 2.	>The algorithm design itself does not appear to be very novel and uses fairly standard statistical estimation techniques.
>
> **Reply:** Thank you for the comment. This paper is the first to study performative prediction under inequality constraints. Compared with conventional constrained optimization problems, performative prediction poses several unique challenges.
>
> The first main challenge is that the data distribution in performative prediction varies with the decision taken, leading to a changing optimization objective, i.e., performative risk. Another challenge is that, in performative prediction, the performative risk remains unknown throughout the optimization process due to the uncertainty of the underlying distribution. Therefore, anticipating performative gradients is a challenging task. Most existing work simplifies this problem by identifying performative stable points. However, performative stability does not necessarily imply performative optimality. In our paper, we aim to find the performative optima, which is more challenging than working on performative stability.
>
> To address these challenges, we design a robust primal-dual framework. Unlike most existing work on stochastic optimization, our framework does not require unbiased gradient approximations. It is worth noting that convergence analysis with a biased approximated gradient is more challenging than the unbiased case. However, we prove that our robust primal-dual framework delivers the same order of performance as the stochastic primal-dual algorithm without performativity. Our technical contributions have been acknowledged and recognized by the reviewers PDku and NMva,  which is evidenced by their comments on our algorithm.
>
>
> **Reply to Questions:**
>
> **1.  Reply to Question 1:** Thank you for the question.  The derivation of Eq. (4) is given below:
>
> Denote by $p_\theta(Z)$ the density of $D(\theta)$. We have
>
> $PR(\theta) = \mathbb{E}_{Z\sim D(\theta)} \ell (\theta; Z)$
>
> $ ~~~~~~~~~~~= \int_Z \ell(\theta; Z) p_\theta(Z) dZ.$
>
>
> Then, $\nabla_\theta PR(\theta) = \int_Z\nabla_\theta \ell\left(\theta; Z\right) p_\theta(Z) + \ell\left(\theta; Z\right) \nabla_\theta p_\theta(Z) dZ$.
>
> Since $\nabla_\theta p_\theta(Z) = \nabla_\theta \exp(\log(p_\theta(Z))) =p_\theta(Z) \nabla_\theta \log(p_\theta(Z))$, we have
>
> $\nabla_\theta PR(\theta)  = \mathbb{E}_{Z\sim D(\theta)} $
>
> $\left[\nabla_\theta \ell(\theta; Z) + \ell(\theta; Z)\nabla_\theta \log(p_\theta(Z))  \right]$.
>
>  **2. Reply to Question 2:** Thank you for the question. As replied to Weakness 1, the proposed robust primal-dual framework can be applied to any distribution families that have effective gradient approximation methods.
>
> As pointed out by the reviewer, the sample complexity for gradient approximation depends on the parametric assumptions on the distribution map $D(\theta)$. For example, [Izzo et al., 2021] assume that $D(\theta)$ is in an exponential family with density $p(Z, f(\theta))$. The corresponding gradient has unknown quantities $\mathbb{E}_{Z\sim D(\theta)} \nabla \ell\left(\theta; Z\right) $, $ f(\theta) $ and $\frac{df}{d\theta }$. [Izzo et al., 2021] demonstrated that approximating these quantities requires $n\geq\mathcal{O}(log(T))$ samples at each iteration. Therefore, the sample complexity of their algorithm is $\mathcal{O}(T(log(T))$, higher than ours.
>
>  **3. Reply to Question 3:**  We appreciate the reviewer for bringing this to our attention. In performative prediction, decisions can actively shape the system being predicted. It considers the feedback loops and interactions between the decisions and the system. In contrast, the basic idea of the prediction problem in 1 and 2 is to incorporate ML predictions in the design of online algorithms to improve their performance. These two approaches are similar in the sense that improving decision-making by estimating unknown quantities. However, several key distinctions set them apart. Firstly, performative prediction involves a dynamic interplay between the decisions made and the underlying system. In contrast, the prediction problem does not influence the system through the decisions taken. Furthermore, in online optimization, the current loss function becomes apparent after the decision has been made, while in performative prediction, the performative risk remains unknown throughout the optimization process. Nevertheless, we believe that research on performative prediction can draw some insights from the prediction problem on incorporating prediction into decision-making. We take it as a promising research direction.
>
>
> **We would be happy to discuss with the reviewer should there be any other concerns/questions. Thank you very much!**

---

### Official Review · Reviewer_qFHm · 2023-07-07

**Soundness:** 3 good
**Presentation:** 4 excellent
**Contribution:** 2 fair
**Rating:** 5
**Confidence:** 2

**Summary:**

The paper studies the problem of performative prediction under inequality constraints and gives an no-regret learning algorithm that obtains $O(\sqrt{T})$ regret and uses $\sqrt{T} + 2T$ samples.

In the problem of performative prediction, the data distribution $Z\sim D(\theta)$ depends on the choice $\theta$ of decision maker, and one formulates it as an optimization problem $\min_{\theta}E_{Z\sim D(\theta)}\ell(\theta, Z)$ and one only has sample access to the distribution $D(\theta)$. The major difference from previous work is that the paper considers a constrained optimization problem, where $\theta$ needs to satisfies certain constraints.

The paper proposes to use primal-dual gradient descent-ascent to solve the problem. Especially, the paper considers location families (where $Z \sim Z_0 + A\theta$ for some unknown matrix $A$) and provides convergence guarantee when the loss function is well-behaved (e.g. strongly convex and smooth).

Numerical experiments are also conducted to verify the practicality of the proposed method.

**Strengths:**

The performative prediction task is a well-studied problem and the constrained one (studied by this paper) is well-motivated. The primal-dual method proposed in this paper is practical and the author also provides regret guarantee under assumptions. The writing is clear and easy to read.

**Weaknesses:**

The technical novelty is limited, the primal dual method (and its robustness to gradient error) is fairly common in the literature.

**Questions:**

The theoretical result mainly focus on the cumulative regret, but as an optimization problem, it is also important to obtain a good decision at the end of $T$ rounds. Do you think taking the average of $\theta_1,\dots,\theta_T$ would be result in a feasible decision with optimality guarantee?

I have a few questions regarding the writing of the paper:

(1)  Algorithm Line 4,6, it should be $A$ instead of $\hat{A}_t$?

(2) Assumption 3, what do you mean by saying the vector valued function $g(\theta)$ is convex?

(3) Line 266, the LHS should be $g_i$ instead of $g_m$, right?

**Limitations:**

.

---

> ### Author Rebuttal · Authors · 2023-08-03
>
> Thank you for your review and for recognizing our contribution to the research field of performative prediction. Our point-to-point responses to concerns on weaknesses and questions are given below:
>
> **Reply to Weaknesses:** Thank you for the comment. The primal-dual method is widely used in the literature as an effective tool for solving constrained optimization problems. However, performative prediction poses several technical challenges that are unique to this field.
>
> The first challenge is that the data distribution in performative prediction varies with the decision taken, leading to a changing optimization objective, i.e., performative risk. Although the current loss function is also unknown in online optimization, it is revealed after the current decision is made. On the other hand, in performative prediction, due to the uncertainty of the underlying distribution, the exact performative risk remains unknown throughout the optimization process. This leads to the second challenge in our problem. Most existing work simplifies this problem by identifying performative stable points that minimize the loss function under the distribution it induces. However, performative stability does not necessarily imply performative optimality.
>
> In our paper, we aim to find the performative optima for the constrained performative prediction problem, which is a more stringent solution concept than performative stability. To solve this problem, we design a robust primal-dual framework that accepts inexact gradient approximations up to a certain accuracy. Our primal-dual framework does not restrict the approximated gradients to be unbiased, offering more flexibility for the gradient approximation design. Convergence analysis with a biased approximated gradient is more challenging than the unbiased case. However, we prove that the proposed primal-dual framework delivers the same order of performance as the stochastic primal-dual algorithm without performativity.
>
> Moreover, our robust primal-dual framework can be applied to any form of distribution families if they have effective gradient approximation methods. Thus, it is more general than the algorithms proposed in two other papers  [Miller et al., 2021] and [Izzo et al., 2021] that focus on the optimality of performative prediction without constraints. Specifically, [Miller et al., 2021] present an algorithm that applies only to the location family, while the algorithm of [Izzo et al., 2021] only works for the exponential family.
>
> Our another technical contribution is the design of an adaptive gradient approximation method for the distribution of the location family. This gradient approximation method attains an $\mathcal{O}(\sqrt{T})$ regret using only $\sqrt{T}+2T$ samples. Lastly, we emphasize that we are the first to study performative prediction under inequality constraints, which is of great significance to the field of performative prediction. Our contributions have been acknowledged and recognized by other reviewers, as evidenced by the citations provided below:
>
> - The reviewer “Puku”: “This paper studies and analyze the optimality of the performative prediction problem under inequality constraints, which is an important yet missing piece in the literature. As far as I know, this is the first paper attempts to provide a solution to this problem. The paper is well-written, easy to follow, and the proposed robust primal-dual method is intuitive. The theoretical analysis is throughout and it also provides empirical study to justify the findings.”
>
> -  The reviewer “NMVA”:  “The proposed primal-dual algorithm is very nice in the sense that (i) the gradient estimators don't need to be unbiased; and (ii) the authors prove a regret bound in terms of the accumulated gradient approximation error. This can be a valuable contribution to the research community.”
>
> -  The reviewer “9 way”: “The motivation behind introducing constraints in performative prediction is strong and justified. The explicit regret guarantees are compelling.”
>
> **Reply to Questions:**
>
> **1. Reply to Question 1:**  Thank you for the question. We agree with the reviewer that the average of the decisions $\boldsymbol
> {\theta}_1,\cdots,\boldsymbol{\theta}_T$ is a decision with an optimality guarantee. This is because, under the assumptions made in our paper, we have proved that the performative risk ${PR}(\boldsymbol{\theta})$ is convex with respect to $\boldsymbol{\theta}$ in Lemma 2.
>
> By applying Jensen's inequality, we obtain $PR\left( \frac{1}{T}\sum_{t=1}^T \boldsymbol\theta_t \right) \leq \frac{1}{T}\sum_{t=1}^T PR(\boldsymbol\theta_t)$. This implies $PR\left( \frac{1}{T}\sum_{t=1}^T \boldsymbol\theta_t \right) - PR(\boldsymbol\theta^*) \leq \frac{1}{T}\sum_{t=1}^T PR(\boldsymbol\theta_t) - PR(\boldsymbol\theta^*) \leq \mathcal{O}\left(\frac{1}{\sqrt{T}}\right)$.
>
> Similarly, under assumption 3,  each entry of the constraint function $\mathbf{g}(\theta)$ is convex. Then, we have
> $\mathbb{E}\left[g_i\left( \frac{1}{T}\sum_{t=1}^T \boldsymbol\theta_t \right) \right]\leq \mathbb{E}\left[ \frac{1}{T}\sum_{t=1}^T g_i(\boldsymbol\theta_t)\right] \leq \mathcal{O}\left(\frac{1}{\sqrt{T}}\right)$, $\forall i\in[m]$.
>
>
> Therefore, the average of the decisions $\boldsymbol
> {\theta}_1,\cdots,\boldsymbol{\theta}_T$ converges asymptotically to the performative optimal point as the iteration number $T$ goes to infinity.
>
>
> **2. Reply to  questions regarding the writing of the paper:**  We thank the reviewer for pointing these out.  We have carefully proofread the manuscript and rectified all the typos. For the question (2), "the vector valued function $\mathbf{g}(\theta)$ is convex" means that each entry of the vector-valued function $\mathbf{g}(\theta)$ is convex.
>
> **We would be happy to discuss with the reviewer should there be any other concerns/questions. Thank you very much!**

---

> > ### Comment · Reviewer_qFHm · 2023-08-20
> >
> > Thank you for explaining the technical challenges. I have raised my score.

---

> > > ### Author Response · Authors · 2023-08-21
> > >
> > > Thank you for recognizing our contributions!

---

### Official Review · Reviewer_PDku · 2023-07-09

**Soundness:** 3 good
**Presentation:** 3 good
**Contribution:** 3 good
**Rating:** 7
**Confidence:** 3

**Summary:**

 This paper studies performative prediction under inequality constraints. In particular, the paper develops a robust primal-dual framework that requires only approximate gradients up to a certain accuracy but achieves the same order of performance as the stationary stochastic primal-dual algorithm even without performativity. Based on this framework, the authors propose an adaptive primal-dual algorithm for location families. The paper also presents the regret analysis and the perform numerical simulations to validate their findings.

**Strengths:**

This paper studies and analyze the optimality of the performative prediction problem under inequality constraints, which is an important yet missing piece in the literature. As far as I know, this is the first paper attempts to provide a solution to this problem. The paper is well-written, easy to follow, and the proposed robust primal-dual method is intuitive. The theoretical analysis is throughout and it also provides empirical study to justify the findings.

**Weaknesses:**

The major weakness of the paper is the assumptions are very strong, and the setting is somewhat restricted. For example, the proposed algorithm only works for a particular family of distribution, namely the location family, and it requires an accurate estimation of all the parameters involved in the computation, which might not be realistic. However, it is also almost unavoidable for the performative prediction setting.

**Questions:**

1. Can the author be more explicit about the dependency on the sample required per iteration? In particular, for line 282, why does the initial sample $n\geq \sqrt{T}$ implies the sum of the expected gradient difference is bounded by $O(\sqrt{T})$?

---

> ### Author Rebuttal · Authors · 2023-08-02
>
> Thank you for your review and for recognizing our contribution to the research field of performative prediction. Our point-to-point responses to concerns on weaknesses and questions are given below:
>
> **Reply to Weaknesses:**
>
> 1) >The major weakness of the paper is the assumptions are very strong.
>
> **Reply:** Thank you for the comment.  We agree with the reviewer that the assumptions made in our paper are relatively strong, although they are standard in the literature of performative prediction and constrained optimization. It may be possible to eliminate some of the assumptions, such as the strong-convexity of the loss function may be removed by utilizing techniques for nonconvex-concave problems [Valkonen, 2021] [Zhao, 2020]. We leave it as a future research direction.
>
> •	Valkonen, Tuomo. "First-order primal–dual methods for nonsmooth non-convex optimisation." Handbook of Mathematical Models and Algorithms in Computer Vision and Imaging: Mathematical Imaging and Vision (2021): 1-42.
>
> •	Zhao, Renbo. "A Primal-Dual Smoothing Framework for Max-Structured Non-Convex Optimization." arXiv preprint arXiv:2003.04375 (2020).
>
>
> 2)	> The setting is somewhat restricted. For example, the proposed algorithm only works for a particular family of distribution, namely the location family. and it requires an accurate estimation of all the parameters involved in the computation, which might not be realistic.  However, it is also almost unavoidable for the performative prediction setting.
>
> **Reply:** Thank you for the comment. The proposed robust primal-dual framework has no restrictions on data distributions but only requires that the gradient approximation is up to a certain accuracy. To be specific, the regret bound of the primal-dual framework is given by
>
> $$ \sum_{t=1}^T \left(\mathbb{E}PR(\theta_t) -   PR(\theta^*)\right) \leq \frac{\gamma \sqrt{T}}{\gamma-a} \left(2R^2 + C^2 + 2L^2 \right) + \frac{\gamma}{\gamma-a}\left(\frac{1}{2a} + \frac{1}{\sqrt{T}} \right) \sum_{t=1}^T\mathbb{E} ||\nabla \widehat{PR}_t(\theta_t)-\nabla PR(\theta_t)||_2^2,$$
>
> where $a \in (0,\gamma)$ is a constant and $\gamma$ is the strongly-convex parameter of the performative risk. Denote by $\mathbf{e}(T) := \sum_{t=1}^T \mathbb{E} ||\nabla \widehat{ PR}_t(\theta_t) - \nabla PR(\theta_t) ||_2^2$. Then, the regret of the robust primal-dual framework is upper bounded by $\mathcal{O}\left(\sqrt{T}+\mathbf{e}(T)\right)$. To achieve an $\mathcal{O}(\sqrt{T})$ regret bound, we only require that the accumulated gradient approximation error $\mathbf{e}(T)\leq\mathcal{O}(\sqrt{T})$, and there is no restriction on the forms of distributions.
>
> In this paper, as an example, we consider distribution maps of the local family, where the performative effect is linear w.r.t. the decision. Our robust primal-dual framework can be applied to other distribution families if they have effective gradient approximation methods. For instance, the exponential family considered in [Izzo et al., 2021] with their gradient approximation method can be directly applied to our robust primal-dual framework.
>
> We thank the reviewer for pointing this out. We have clarified the generalizability of our robust primal-dual framework to other distributions in the revised manuscript.
>
> **Reply to Questions:**
>
> 1)  > Can the author be more explicit about the dependency on the sample required per iteration?
>
> **Reply:** Thank you for the question. First, at the start of our algorithm,  we draw $\sqrt{T}$ samples from $\mathcal D(\mathbf{0})$. These samples are used to approximate the base distribution $\mathcal{D}_0$ throughout the algorithm iteration. Then, at each iteration, we require two samples $Z_t \sim \mathcal{D}(\theta_t)$ and $Z^\prime_t \sim \mathcal{D}(\theta_t+u_t )$, where $\theta_t$ is the current decision and $u_t$ is an injected noise. These two samples are for constructing the online least-square objective to update the estimate of the parameter $\mathbf{A}$. Overall, our algorithm requires $\sqrt{T} + 2T$ samples.
>
> 2)  >In particular, for line 282, why does the initial sample $n\geq \sqrt{T}$ implies the sum of the expected gradient difference is bounded by $\mathcal{O}(\sqrt{T})$?
>
> **Reply:** Thank you for the question. From Lemma 4, the error of our gradient approximation is upper bounded by:
>
>  $$ \sum_{t=1}^T \mathbb{E} ||\nabla \widehat{PR}_t(\theta_t) - \nabla  PR(\theta_t) ||_2^2
>     \leq \frac{2T\sigma^2}{n} + \frac{4}{n} \left(2L_Z^2 + \beta^2 R^2 (1 + 2\sigma(\mathbf{A}))  \right)\bar{\alpha}\ln(T) = \mathcal{O}\left( \frac{T}{n} + \ln{T}\right), $$
>
>
> where  $\bar{\alpha}:=\max\left(\left(1+\frac{2\kappa_3}{\kappa_1} \right) ||\hat{\mathbf{A}}_1 - \mathbf{A}||_F^2 ,  \frac{8\kappa_2~tr(\Sigma)}{\kappa_1^2}\right)$.
>
>   Since $ \ln(T) \leq \sqrt{T}$, when the number of initial samples $n \geq \sqrt{T}$, we have $\mathcal{O}\left( \frac{T}{n} + \ln{T}\right) \leq \mathcal{O}( \sqrt{T}) $ and thus  the expected gradient difference $ \sum_{t=1}^T \mathbb{E} ||\nabla \widehat{ PR}_t(\theta_t) - \nabla  PR(\theta_t) ||_2^2 \leq \mathcal{O}( \sqrt{T}) $.
>
> **We would be happy to discuss with the reviewer should there be any other concerns/questions. Thank you very much!**

---

> > ### Comment · Reviewer_PDku · 2023-08-14
> >
> > Thank you for the rebuttal. My questions are answered and my evaluation remains the same.

---

> > > ### Author Response · Authors · 2023-08-15
> > >
> > > Thank you for recognizing our contributions!

---

### Official Review · Reviewer_NMva · 2023-07-20

**Soundness:** 4 excellent
**Presentation:** 4 excellent
**Contribution:** 3 good
**Rating:** 7
**Confidence:** 3

**Summary:**

This paper studies the problem of performative prediction under inequality constraints. The authors propose an adaptive, robust primal-dual algorithm that achieves sublinear regret and constraint violations, where the regret is with respect to the performative optima (instead of just a performative stable point). The algorithm is robust in the sense that it only requires approximate gradients up to a certain accuracy level and doesn't require the estimated gradients to be unbiased.

**Strengths:**

* The main paper is very clearly written. The remarks around lemmas and theorems are helpful in interpreting the formal results.

* The proposed primal-dual algorithm is very nice in the sense that (i) the gradient estimators don't need to be unbiased; and (ii) the authors prove a regret bound in terms of the accumulated gradient approximation error. This can be a valuable contribution to the research community.

* The theoretical results are supplemented by good experimental results on multi-task linear regression and multi-asset portfolio.

**Weaknesses:**

* The model assumes the learner can query the distribution for samples (lines 204 - 213). In particular, the learner can query the performative risk at $\theta_t$ and $\theta_t + u_t$. This is a strong assumption and should be stated more explicitly. What if the learner cannot query the distribution and only observes a single sample (or, say, $k$ samples) in each round depending on $\theta_t$? This is more similar to the setting of Jagdeesan et al.

* I acknowledge that Assumption 1 is standard in the literature on performative prediction, but I find the strong-convexity assumption quite strong.

* There is no discussion on lower bounds for the sample complexity.

* The bound seems to have a factor of $d$ (the dimension). It is unclear to me whether this can be improved and how this compares to existing results in the performative prediction literature and results on performative regret in the unconstrained setting. (See one my questions below.)

**Questions:**

* I really like example 1 (multi-asset portfolio). However, I have one question - can you elaborate on how "excessive investment in a particular asset can lead to a declination of its rate of return"?

* Do you have thoughts on whether the sample complexity of $\sqrt{T} + 2 T$ is tight?

* Can you provide some intuition for why the third part of Assumption 5 is needed?

* What is the dependence of the bounds on the dimension $d$? For example, for the case of Gaussian noise, $\kappa_2 = d$ and $\kappa_3 = 3d$ (line 250) and in Lemma 4 this can result in a factor as large as $d$ (through the definition of $\bar{\alpha}$. Can this dependence be improved? Do existing results in the performative prediction literature (and results on performative regret in the unconstrained setting) have a similar dependence?

* (This is a comment, not a question.) It would help to include some proof sketches in the main paper. (I have not checked the proofs in the appendix.)

**Limitations:**

Limitations: Assumptions are clearly stated in Section 4. (The assumption that one can query the distribution could be stated more explicitly.)
Negative societal impact: N/A.

---

> ### Author Rebuttal · Authors · 2023-08-05
>
> Thank you for your review and for recognizing our contribution to the research field of performative prediction. Our point-to-point responses to concerns on weaknesses and questions are given below:
>
> **Reply to Weaknesses:**
>
> **1. Reply to Weakness 1:**  Thank you for the comment. Our algorithm only needs to query the data distribution $D(\theta)$, not the performative risk. Specifically,  we first draw $n$ samples from the distribution $D (\mathbf{0})$ at the start of our algorithm. Then, at each iteration, we requires two samples $Z_t \sim D\left(\theta_t\right)$ and $Z^\prime_t \sim D\left(\theta_t+u_t \right)$, where $\theta_t$ is the current decision and $u_t$ is an injected noise. This setting aligns with [Jagdeesan et al., 2022] and is standard in the literature of performative prediction. We apologize for any potential misunderstanding in our paper, and we have clarified this in the revised version.
>
>
> **2. Reply to Weakness 2:**  Thank you for the comment. We agree with the reviewer that the strong-convexity assumption on the loss function is relatively strong, although it is standard in performative prediction. To enforce strong-convexity, a common approach is to add a regularizer $||\theta||_2^2$ on the loss function,  as adopted by [Narang et al., 2022] and [Mandal et al., 2022]. It may be possible to eliminate the strong-convexity assumption in constrained performative prediction by utilizing techniques for nonconvex-concave problems. We leave it as a future research direction.
>
> **3. Reply to Weakness 3:**  Thank you for the comment. We have added a remark on the sample complexity of our algorithm in Section 4 of our revised manuscript, as cited below:
>
>  >"Over the time horizon $T$, Algorithm 1 requires a total of $\sqrt{T}+2T$ samples. In these samples, $\sqrt{T}$ samples are used to approximate the expectation over $Z_{0}$. Additionally, $2$ samples are required at each iteration to construct the online least-square objective.  "
>
>
> **4. Reply to Weakness 4:**  Thank you for the comment.  Please find the answer to this comment in Reply to Question 4.
>
> **Reply to Questions:**
>
> **1. Reply to Question 1:**  Thank you for the question. Excessive investment in a single asset can lead to an overvaluation of that asset relative to its underlying fundamentals. If the asset's price becomes overvalued, it may deter other investors from buying into the asset, reducing its demand in the market. This reduction in demand can lead to a decline in the asset's price as investors try to sell their holdings to avoid potential losses, further exacerbating the decline.
>
> **2. Reply to Question 2:**  Thank you for the question. Our algorithm achieves the same order of sample complexity, i.e., $\mathcal{O}(T)$, as the stochastic algorithm that performs gradient updates based on a single sample in each iteration. Moreover, compared to three other papers ([Miller et al., 2021], [Jagadeesan et al., 2022], and [Izzo et al., 2021]) on performative optimality, the sample complexity of our algorithm is of the same order as that of [Miller et al., 2021], while is lower than that of [Jagadeesan et al., 2022] and [Izzo et al., 2021], whose sample complexity are both $\mathcal{O}(T\log(T))$.
>
> **3. Reply to Question 3:**  Thank you for the question. In online least-squares, the update rule of the parameter estimate is $\hat{\mathbf{A}}_{t+1} = \hat{\mathbf{A}}_t - \zeta_t(\hat{\mathbf{A}}_t \mathbf{u}_t - (Z^\prime_t - Z_t))\mathbf{u}_t^\top$, where $\zeta_t$ is the stepsize. To evaluate the parameter estimation error, we compute
>
> $$||\hat{\mathbf{A}}_{t+1} - \mathbf{A}||_F^2 =||\hat{\mathbf{A}}_t  - \mathbf{A}  - \zeta_t(\hat{\mathbf{A}}_t \mathbf{u}_t - (Z^\prime_t - Z_t))\mathbf{u}_t^\top||_F^2.$$
>
> Decomposing the above equation yields a term related to $||\mathbf{u}_t||_2^2 \mathbf{u}_t \mathbf{u}_t^\top$. Thus, we need the third part of Assumption 5.
>
>
> **4. Reply to Question 4:**  Thank you for the question. Our regret bound is given by $\mathcal{O}(d\sqrt{T})$ if we take the decision dimension $d$ into consideration. The factor $d$ arises in our regret bound due to the parameter estimation error. We cannot remove $d$ from our bound because the estimation error of the online least-squares is proportional to the number of parameters that need to be estimated.
>
> In the literature, the performance bounds of other papers on performative optimality also depend on the decision dimension $d$ by $\mathcal{O}(d)$, including [Miller et al., 2021], [Jagadeesan et al., 2022], [Izzo et al., 2021], and [Narang et al., 2022].
>
> **5. Reply to Question 5:**  Thank you for the comment. We have added a proof sketch at the start of the Section 4. Specifically, in our analysis, we first provide the convergence result of the robust primal-dual framework, which depends on the accumulated gradient approximation error. Then, we bound the error of gradient approximation by our adaptive algorithm. With these results, the convergence bounds of the proposed adaptive primal-dual algorithm are derived.
>
> **We would be happy to discuss with the reviewer should there be any other concerns/questions. Thank you very much!**

---

> > ### Comment · Reviewer_NMva · 2023-08-10
> >
> > Thank you for your response. I will increase my rating by one point.

---

> > > ### Author Response · Authors · 2023-08-11
> > > **Reply to Reviewer NMva**
> > >
> > > Thank you for recognizing our contributions!

---

### Decision · Program_Chairs · 2023-09-21

**Decision:**

Accept (poster)

**Comment:**

This paper generated quite a bit of discussion bother during the author-reviewer conversations and after. Overall, it had two scores of 7, while the support from other reviewers was lukewarm. This state along with the average score puts the paper near the borderline of the acceptance-rejection region for the conference.

On one hand, the reviewers appreciate that this work addresses a missing piece in the literature on performative prediction and that it is written well. On the other hand, even staunch supporters point out that the restrictions here are severe, e.g., operating only a particular family of linearly parametrized distributions and strong convexity. In totality, I agree that the authors study a new variation of the performative prediction problem and the work could be valuable to the community, and I also concur also with Reviewer 9Way that the proposal here combines existing techniques (e.g., dual descent is a common approach in online convex opt with approximately satisfied functional constraints), thereby offering possibly limited technical novelty.

I am tentatively recommending to accept the paper, although I do encourage the authors to see if they can broaden the scope of applicability of their results (beyond linear parametrized distributions) even at the cost of some dip in clarity.